# Learning with Kan Extensions

## Abstract

A common problem in machine learning is "use this function defined over this small set to generate predictions over that larger set." Extrapolation, interpolation, statistical inference and forecasting all reduce to this problem. The Kan extension is a powerful tool in category theory that generalizes this notion. In this work we explore applications of the Kan extension to machine learning problems. We begin by deriving a simple classification algorithm as a Kan extension and experimenting with this algorithm on real data. Next, we use the Kan extension to derive a procedure for learning clustering algorithms from labels and explore the performance of this procedure on real data.

Although the Kan extension is usually defined in terms of categories and functors, this paper assumes no knowledge of category theory. We hope this will enable a wider audience to learn more about this powerful mathematical tool.

## 1 Introduction

A popular slogan in category theoretic circles, popularized by Saunders Mac Lane, is: "all concepts are Kan extensions" (Mac Lane, 1971). While Mac Lane was partially referring to the fundamental way in which many elementary category theoretic structures can be formulated as Kan extensions, there are many applied areas that have Kan extension structure lying beneath the surface as well.

Casting a problem as a Kan extension can unveil hidden structure and suggest new avenues for exploration. In this paper we aim to demonstrate what Kan extensions, and applied category theory more broadly, can offer to machine learning researchers. We hope that our work will inspire more researchers to explore this direction.

As a machine learning researcher it may be easiest to think of the Kan extension as tool for extrapolation. We can use the Kan extension to expand a function over a small set to a similar function over a larger set. However, the Kan extension perspective on extrapolation is fundamentally different from traditional machine learning perspectives. Intuitively, traditional perspectives focus on means and sums whereas the Kan extension perspective focuses on minimums and maximums. That is, a traditional machine learning algorithm may try to extrapolate from data in a way that minimizes the total observed error. In contrast, an algorithm derived from the Kan extension may try to solve a problem like "minimize false positives subject to no false negatives on some set".

In this paper we explore the ramifications of this difference across supervised and unsupervised learning applications. To do this, we cast basic machine learning problems in category theoretic language, apply the Kan extension, translate the result back to machine learning language, and study the behavior of the resulting algorithms.

First, we derive a simple classification algorithm as a Kan extension and demonstrate experimentally that this algorithm can learn to classify images. Next, we use Kan extensions to derive a novel method for learning a clustering algorithm from labeled data and demonstrate experimentally that this method can learn to cluster images. All code is available on GitHub.

For interested readers we include two additional examples of how Kan extensions can be applied to machine learning in the Appendix. In Section A.2 we explore the structure of meta-supervised learning and use Kan

extensions to derive supervised learning algorithms from sets of labeled datasets and trained functions. In Section A.3 we use Kan extensions to characterize the process of approximating a complex function with a simpler minimum description length (MDL) function.

## 2 Preliminaries

The foundational structures that we use in this paper are preorders and monotonic maps.

**Definition 2.1.** *A preorder $(P, \leq)$ is a tuple of a set of objects $P$ and a reflexive, transitive relation $\leq$ on $P$.*

**Definition 2.2.** *A monotonic map $f : (P_1, \leq_1) \to (P_2, \leq_2)$ from the preorder $(P_1, \leq_1)$ to the preorder $(P_2, \leq_2)$ is an order-preserving function from $P_1$ to $P_2$. That is, for any $x, y \in P_1$ where $x \leq_1 y$ we have $f(x) \leq_2 f(y)$.*

For example, consider the preorder $(\mathbb{R}^n, \leq_{\|\|})$ where for $v, u \in \mathbb{R}^n$ we have $v \leq_{\|\|} u$ when $\|v\| \leq \|u\|$. Consider also the preorder $(\mathbb{R}^n, \leq_\forall)$ where $v \leq_\forall u$ when $\forall_{i=1\cdots n}|v_i| \leq |u_i|$. The identity function $id : (\mathbb{R}^n, \leq_\forall) \to (\mathbb{R}^n, \leq_{\|\|})$ is a monotonic map, but the identity function $id : (\mathbb{R}^n, \leq_{\|\|}) \to (\mathbb{R}^n, \leq_\forall)$ is not a monotonic map.

**Definition 2.3.** *$(P, \leq)$ is a discrete preorder if $p_1 \leq p_2$ in $P$ implies $p_1 = p_2$*

**Definition 2.4.** *$(P_1, \leq_1)$ is a subpreorder of the preorder $(P_2, \leq_2)$ if $P_1 \subseteq P_2$ and the identity function $id : (P_1, \leq_1) \to (P_2, \leq_2)$ is a monotonic map.*

For example, $(\mathbb{R}^n, \leq_\forall)$ is a subpreorder of $(\mathbb{R}^n, \leq_{\|\|})$. As another example, consider the preorder $(U^n, \leq_\forall)$ where $U^n$ is the set of unit-norm vectors in $\mathbb{R}^n$. $(U^n, \leq_\forall)$ is a subpreorder of $(\mathbb{R}^n, \leq_\forall)$.

In order to keep notation simple we will use bold characters like $\mathbf{A}$ to represent the preorder $(Ob(\mathbf{A}), \leq_\mathbf{A})$. In addition, given two monotonic maps $f_1, f_2 : \mathbf{A} \to \mathbf{B}$ we will write $f_1 \leq f_2$ to indicate that for all $a \in \mathbf{A}$ we have $f_1(a) \leq f_2(a)$.

### 2.1 Kan Extensions

Now suppose we have three preorders $\mathbf{A}, \mathbf{B}, \mathbf{C}$ and two monotonic maps $G : \mathbf{A} \to \mathbf{B}, K : \mathbf{A} \to \mathbf{C}$ and we would like to derive the "best" monotonic map $F : \mathbf{B} \to \mathbf{C}$:

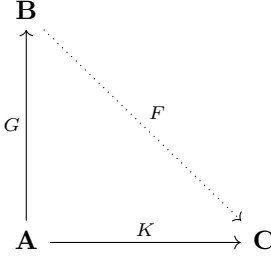

There are two canonical ways that we can do this.

**Definition 2.5.** *The left Kan extension of $K : \mathbf{A} \to \mathbf{C}$ along $G : \mathbf{A} \to \mathbf{B}$ is the minimal monotonic map $Lan_G K : \mathbf{B} \to \mathbf{C}$ such that $K \leq (Lan_G K \circ G)$.*

That is, for any other monotonic map $m : \mathbf{B} \to \mathbf{C}$ such that $K \leq (m \circ G)$ we have $Lan_G K \leq m$.

**Definition 2.6.** *The right Kan extension of $K : \mathbf{A} \to \mathbf{C}$ along $G : \mathbf{A} \to \mathbf{B}$ is the maximal monotonic map $Ran_G K : \mathbf{B} \to \mathbf{C}$ such that $(Ran_G K \circ G) \leq K$.*

That is, for any other monotonic map $m : \mathbf{B} \to \mathbf{C}$ such that $(m \circ G) \leq K$ we have $m \leq Ran_G K$.

If $G : \mathbf{A} \hookrightarrow \mathbf{B}$ is the inclusion map then the Kan extensions of $K$ along $G$ are interpolations or extrapolations of $K$ from $\mathbf{A}$ to all of $\mathbf{B}$. For example, suppose we want to interpolate a monotonic function $K : \mathbb{Z} \to \mathbb{R}$ to a monotonic function $F : \mathbb{R} \to \mathbb{R}$ such that $F \circ G = K$ where $G : \mathbb{Z} \hookrightarrow \mathbb{R}$ is the inclusion map.

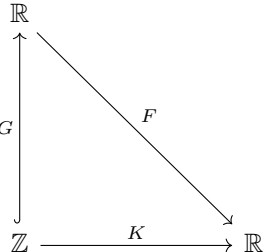

We have that $Lan_G K : \mathbb{R} \to \mathbb{R}$ is simply $K \circ floor$ and $Ran_G K : \mathbb{R} \to \mathbb{R}$ is simply $K \circ ceil$, where $floor, ceil$ are the rounding down and rounding up functions respectively.

In this paper we explore a few applications of Kan extensions to machine learning. In each of these applications we first define preorders $\mathbf{A}, \mathbf{B}, \mathbf{C}$ and a monotonic function $K : \mathbf{A} \to \mathbf{C}$ such that $\mathbf{A}$ is a subpreorder of $\mathbf{B}$ and $G : \mathbf{A} \hookrightarrow \mathbf{B}$ is the inclusion map. Then, we take the left and right Kan extensions $Lan_G K, Ran_G K$ of $K$ along $G$ and study their behavior.

## 3 Classification

We start with a simple application of Kan extensions to supervised learning. Suppose that $\mathbf{I}$ is a preorder, $\mathbf{I}' \subseteq \mathbf{I}$ is a subpreorder of $\mathbf{I}$, and $\{false, true\}$ is the two element preorder where $false < true$. Suppose also that $K : \mathbf{I}' \to \{false, true\}$ is a mapping into $\{false, true\}$ and we would like to learn a monotonic function $F : \mathbf{I} \to \{false, true\}$ that approximates $K$ on $\mathbf{I}'$. That is, $K$ defines a finite training set of points $S = \{(x, K(x)) \mid x \in \mathbf{I}'\}$ from which we wish to learn a monotonic function $F : \mathbf{I} \to \{false, true\}$. Of course, it may not be possible to find a monotonic function that agrees with $K$ on all the points in $\mathbf{I}'$.

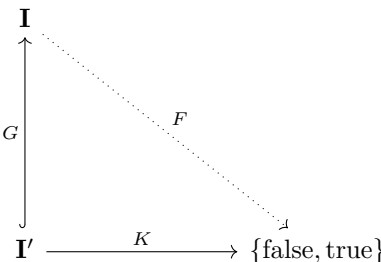

We can solve this problem with the left and right Kan extensions of $K$ along the inclusion map $G : \mathbf{I}' \hookrightarrow \mathbf{I}$.

**Proposition 3.1.** *The left and right Kan extensions of $K : \mathbf{I}' \to \{false, true\}$ along the inclusion map $G : \mathbf{I}' \hookrightarrow \mathbf{I}$ are respectively:*

$$Lan_G K : \mathbf{I} \to \{false, true\} \qquad Ran_G K : \mathbf{I} \to \{false, true\}$$

$$Lan_G K(x) = \begin{cases} true & \exists x' \in \mathbf{I}', x' \leq x, K(x') = true \\ false & else \end{cases}$$

$$Ran_G K(x) = \begin{cases} false & \exists x' \in \mathbf{I}', x \leq x', K(x') = false \\ true & else \end{cases}$$

*(Proof in Appendix A.1.1)*

In the extreme case that $Ob(\mathbf{I}') = \emptyset$, for $x \in \mathbf{I}$ we have that:

$$Lan_G K(x) = \left( \begin{cases} true & \exists x' \in \mathbf{I}', x' \leq x, K(x') = true \\ false & else \end{cases} \right) = false$$

$$Ran_G K(x) = \left( \begin{cases} false & \exists x' \in \mathbf{I}', x \leq x', K(x') = false \\ true & else \end{cases} \right) = true$$

Similarly, in the extreme case that $Ob(\mathbf{I'}) = Ob(\mathbf{I})$ we have by the monotonicity of $K$ that for $x \in \mathbf{I}$ both of the following hold if and only if $K(x) = \text{true}$.

$$\exists x' \in \mathbf{I'}, x' \leq x, K(x') = \text{true} \qquad \nexists x' \in \mathbf{I'}, x \leq x', K(x') = \text{false}$$

Therefore in this extreme case we have $Lan_G K(x) = Ran_G K(x) = K(x)$.

Now suppose that $\mathbf{I'}$ contains at least one $x'$ such that $K(x') = \text{true}$ and at least one $x'$ such that $K(x') = \text{false}$. In this case $Lan_G K$ and $Ran_G K$ split $\mathbf{I}$ into three regions: a region where both map all points to false, a region where both map all points to true, and a disagreement region. Note that $Ran_G K$ has no false positives on $\mathbf{I'}$ and $Lan_G K$ has no false negatives on $\mathbf{I'}$.

For example, suppose $\mathbf{I} = \mathbb{R}, \mathbf{I'} = \{1, 2, 3, 4\}$ and we have:

$$K(1) = \text{false} \qquad K(2) = \text{false} \qquad K(3) = \text{true} \qquad K(4) = \text{true}$$

Then we have that:

$$Lan_G K(x) = \left( \begin{cases} \text{true} & \exists x' \in \mathbf{I'}, x' \leq x, K(x') = \text{true} \\ \text{false} & \text{else} \end{cases} \right) = \left( \begin{cases} \text{true} & x \geq 3 \\ \text{false} & \text{else} \end{cases} \right)$$

$$Ran_G K(x) = \left( \begin{cases} \text{false} & \exists x' \in \mathbf{I'}, x \leq x', K(x') = \text{false} \\ \text{true} & \text{else} \end{cases} \right) = \left( \begin{cases} \text{true} & x > 2 \\ \text{false} & \text{else} \end{cases} \right)$$

In this case the disagreement region for $Lan_G K, Ran_G K$ is $(2, 3)$ and for any $x \in (2, 3)$ we have $Lan_G K(x) < Ran_G K(x)$.

As another example, suppose $\mathbf{I} = \mathbb{R}, \mathbf{I'} = \{5, 6, 7, 8\}$ and we have:

$$K(5) = \text{false} \qquad K(6) = \text{true} \qquad K(7) = \text{false} \qquad K(8) = \text{true}$$

Then we have that:

$$Lan_G K(x) = \left( \begin{cases} \text{true} & \exists x' \in \mathbf{I'}, x' \leq x, K(x') = \text{true} \\ \text{false} & \text{else} \end{cases} \right) = \left( \begin{cases} \text{true} & x \geq 6 \\ \text{false} & \text{else} \end{cases} \right)$$

$$Ran_G K(x) = \left( \begin{cases} \text{false} & \exists x' \in \mathbf{I'}, x \leq x', K(x') = \text{false} \\ \text{true} & \text{else} \end{cases} \right) = \left( \begin{cases} \text{true} & x > 7 \\ \text{false} & \text{else} \end{cases} \right)$$

In this case the disagreement region for $Lan_G K, Ran_G K$ is $[6, 7]$ and for any $x \in [6, 7]$ we have $Ran_G K(x) < Lan_G K(x)$.

While this approach is effective for learning very simple mappings, there are many choices of $K$ for which $Lan_G K, Ran_G K$ do not approximate $K$ particularly well on $\mathbf{I'}$ and therefore the disagreement region is large. In such a situation we can use a similar strategy to the one leveraged by kernel methods (Hofmann et al., 2008) and transform $\mathbf{I}$ to minimize the size of the disagreement region.

That is, we choose a preorder $\mathbf{I^*}$ and transformation $f : \mathbf{I} \to \mathbf{I^*}$ such that the size of the disagreement region for $Lan_{f \circ G} K \circ f, Ran_{f \circ G} K \circ f$ is minimized.

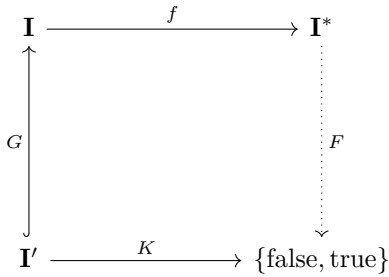

For example, if $\mathbf{I}^* = \mathbb{R}^a$ we can choose $f$ to minimize the following loss:

**Definition 3.2.** *Suppose we have a set $\mathbf{I}' \subseteq \mathbf{I}$ and function $K : \mathbf{I}' \to \{false, true\}$ such that:*

$$\exists x', x'' \in \mathbf{I}', K(x') = true, K(x'') = false$$

*Then the ordering loss $l$ maps a function $f : \mathbf{I} \to \mathbb{R}^a$ to an approximation of the size of the disagreement region for $Lan_{f \circ G} K \circ f, Ran_{f \circ G} K \circ f$. Formally, we define the ordering loss $l$ to be:*

$$l : (\mathbf{I} \to \mathbb{R}^a) \to \mathbb{R}$$

$$l(f) = \sum_{i \leq a} \max(0, \ \max\{f(x)[i] \ | \ x \in \mathbf{I}', K(x) = false\} -$$

$$\min\{f(x)[i] \ | \ x \in \mathbf{I}', K(x) = true\})$$

*where $f(x)[i]$ is the ith component of the vector $f(x)[i] \in \mathbb{R}^a$.*

We can show that minimizing the ordering loss $l$ will also minimize the size of the disagreement region:

**Proposition 3.3.** *The ordering loss $l$ (Definition 3.2) is nonnegative and is only equal to 0 when $\forall x \in \mathbf{I}'$ we have:*

$$K(x) = (Lan_{f \circ G} K \circ f)(x) = (Ran_{f \circ G} K \circ f)(x)$$

*Proof.* First note that $l$ must be nonnegative since each term can be expressed as $\max(0, \_)$. Next, suppose that $l(f) = 0$. Then it must be that for any $x_0, x_1 \in \mathbf{I}'$ such that $K(x_0) = false, K(x_1) = true$ we have that $f(x_0) \leq f(x_1)$. As a result, for any $x \in \mathbf{I}'$ there can only exist some $x' \in \mathbf{I}'$ where $f(x) \leq f(x'), K(x') = false$ when $K(x) = false$. Similarly, there can only exist some $x' \in \mathbf{I}'$ where $f(x') \leq f(x), K(x') = true$ when $K(x) = true$. Therefore:

$$K(x) = (Lan_{f \circ G} K \circ f)(x) = (Ran_{f \circ G} K \circ f)(x)$$

$\square$

It is relatively straightforward to minimize the ordering loss with an optimizer like subgradient descent (Boyd et al., 2003). For example, we can implement the ordering loss in Tensorflow (Abadi et al., 2015) as follows:

```python
import numpy as np
import tensorflow as tf

def get_ordering_loss(model, X, y):
    # model: Tensorflow sequential model
    # X: 2D numpy float array in which each row is a feature vector
    # y: 1D numpy boolean array of labels
    X_false = X[np.logical_not(np.array(y, dtype=bool))]
    false_preds = tf.transpose(model(X_false))

    X_true = X[np.array(y, dtype=bool)]
    true_preds = tf.transpose(model(X_true))

    return tf.reduce_sum(tf.math.maximum(0,
        tf.math.reduce_max(false_preds), axis=1) -
        tf.math.reduce_min(true_preds), axis=1)
```

| Model | Dataset | True Positive Rate | True Negative Rate |
|---|---|---|---|
| Left Kan Classifier | Training | 1.000 ($\pm$0.000) | 0.612 ($\pm$0.042) |
| Right Kan Classifier | Training | 0.705 ($\pm$0.035) | 1.000 ($\pm$0.000) |
| Left Kan Classifier | Testing | 0.815 ($\pm$0.020) | 0.593 ($\pm$0.044) |
| Right Kan Classifier | Testing | 0.691 ($\pm$0.044) | 0.837 ($\pm$0.026) |

**Table 1:** *True positive rate and true negative rate of the left Kan classifier $Lan_{f \circ G} K \circ f$ and the right Kan classifier $Ran_{f \circ G} K \circ f$ where $f$ is a linear map trained to minimize the ordering loss $l(f)$ (Definition 3.2) on the Fashion-MNIST "T-shirt" vs "shirt" task (Xiao et al., 2017). We run a bootstrap experiment by repeatedly selecting 9000 training samples and 1000 testing samples, running the training procedure, and computing true positive rate and true negative rate metrics. Mean and two standard error confidence bounds from 10 such bootstrap iterations are shown.*

In Table 1 we demonstrate that we can use this strategy to distinguish between the "T-shirt" (false) and "shirt" (true) categories in the Fashion MNIST dataset (Xiao et al., 2017). Samples in this dataset have 784 features (pixels), so we train a simple linear model $f : \mathbb{R}^{784} \to \mathbb{R}^{10}$ with Adam (Kingma & Ba, 2014) to minimize the ordering loss $l(f)$ over a training set that contains 90% of samples in the dataset. We then evaluate the performance of the left Kan classifer $Lan_{f \circ G} K \circ f$ and the right Kan classifier $Ran_{f \circ G} K \circ f$ over both this training set and a testing set that contains the remaining 10% of the dataset. We look at two metrics over both sets: the true positive rate and the true negative rate. Recall that the true positive rate of a classifier is the proportion of all true samples which the classifier correctly labels as true and the true negative rate of a classifier is the proportion of all false samples which the classifier correctly labels as false.

As we would expect from the definition of Kan extensions, the left Kan classifier $Lan_{f \circ G} K \circ f$ has no false negatives and the right Kan classifier $Ran_{f \circ G} K \circ f$ has no false positives on the training set. The metrics on the testing set are in line with our expectations as well: the left Kan classifier has a higher true positive rate and the right Kan classifier has a higher true negative rate.

## 4 Clustering with Supervision

Clustering algorithms allow us to group points in a dataset together based on some notion of similarity between them. Formally, we can consider a clustering algorithm as mapping a finite metric space $(X, d_X)$ to a partition of $X$.

In most applications of clustering the points in the metric space $(X, d_X)$ are grouped together based solely on the distances between the points and the rules embedded within the clustering algorithm itself. This is an unsupervised clustering strategy since no labels or supervision influence the algorithm output. For example, agglomerative clustering algorithms like HDBSCAN (McInnes & Healy, 2017) and single linkage partition points in $X$ based on graphs formed from the points (vertices) and distances (edges) in $(X, d_X)$.

However, there are some circumstances under which we have a few ground truth examples of pre-clustered training datasets and want to learn an algorithm that can cluster new data as similarly as possible to these ground truth examples. We can define the supervised clustering problem as follows. Given a collection of tuples

$$S = \{(X_1, d_{X_1}, \mathbb{P}_{X_1}), (X_2, d_{X_2}, \mathbb{P}_{X_2}), \cdots, (X_n, d_n, \mathbb{P}_{X_n})\}$$

where each $(X_i, d_{X_i})$ is a finite metric space and $\mathbb{P}_{X_i}$ is a partition of $X_i$, we would like to learn a general function $f$ that maps a finite metric space $(X, d_X)$ to a partition $\mathbb{P}_X$ of $X$ such that for each $(X_i, d_{X_i}, \mathbb{P}_{X_i}) \in S$ the difference between $f(X_i, d_{X_i})$ and $\mathbb{P}_{X_i}$ is small.

In order to frame this objective in terms of Kan extensions we will first construct our preorder of metric spaces.

**Definition 4.1.** *A nonexpansive map from the metric space $(X, d_X)$ to the metric space $(Y, d_Y)$ is a function $f : X \to Y$ such that for $x_1, x_2 \in X$ we have:*

$$d_Y(f(x_1), f(x_2)) \leq d_X(x_1, x_2)$$

**Definition 4.2.** *In the preorder $\mathbf{Met}_{id}$ the set $Ob(\mathbf{Met}_{id})$ consists of all metric spaces $(X, d_X)$ and $(X, d_X) \leq_{\mathbf{Met}_{id}} (Y, d_Y)$ when $X \subseteq Y$ and the inclusion map $\iota : (X, d_X) \hookrightarrow (Y, d_Y)$ is nonexpansive.*

We can represent a clustering of a set $X$ with a partition $\mathbb{P}_X$ of that set. We can now construct our preorder of partitions.

**Definition 4.3.** *Consider the tuples $(X, \mathbb{P}_X), (Y, \mathbb{P}_Y)$ where $\mathbb{P}_X$ is a partition of $X$ and $\mathbb{P}_Y$ is a partition of $Y$. Then a consistent map $f : (X, \mathbb{P}_X) \to (Y, \mathbb{P}_Y)$ is a function $f : X \to Y$ such that for any set $S_X \in \mathbb{P}_X$ there exists some set $S_Y \in \mathbb{P}_Y$ such that $f(S_X) \subseteq S_Y$.*

**Definition 4.4.** *In the preorder $\mathbf{Part}_{id}$ the set $Ob(\mathbf{Part}_{id})$ consists of all partitions $(X, \mathbb{P}_X)$ and $(X, \mathbb{P}_X) \leq_{\mathbf{Part}_{id}} (Y, \mathbb{P}_Y)$ when $X \subseteq Y$ and the inclusion map $\iota : (X, \mathbb{P}_X) \hookrightarrow (Y, \mathbb{P}_Y)$ is consistent.*

We need one more condition to corral our definition of a clustering map.

**Definition 4.5.** *We say that a monotonic map $f$ from a subpreorder of $\mathbf{Met}_{id}$ to a subpreorder of $\mathbf{Part}_{id}$ is well-behaved if for all $(X, d_X)$ in the domain of $f$ we have that $f(X, d_X) = (X, \mathbb{P}_X)$ for some partition $\mathbb{P}_X$ of $X$.*

Intuitively a well-behaved monotonic map from $\mathbf{Met}_{id}$ to $\mathbf{Part}_{id}$ acts as the identity on underlying sets.

Now given a subpreorder $\mathbf{D} \subseteq \mathbf{Met}_{id}$, a discrete preorder $\mathbf{T} \subseteq \mathbf{D}$, and a well-behaved monotonic map $K : \mathbf{T} \to \mathbf{Part}_{id}$, our goal is to find the best well-behaved monotonic map $F : \mathbf{D} \to \mathbf{Part}_{id}$ such that $F \circ G = K$ where $G : \mathbf{T} \hookrightarrow \mathbf{D}$ is the inclusion map.

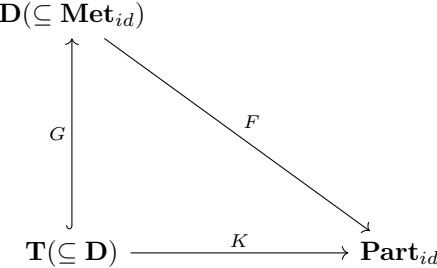

Intuitively, $Ob(\mathbf{T})$ is the set of unlabelled training samples, $K$ defines the labels on these training samples, and $Ob(\mathbf{D})$ is the set of testing samples.

We would like to use the Kan extensions of $K$ along $G$ to find this best clustering map. However, these Kan extensions are not guaranteed to be well-behaved. For example, consider the case in which $\mathbf{T}$ is the discrete preorder that contains the single-element metric space as its only object and $\mathbf{D}$ is the discrete preorder that contains two objects: the single-element metric space and $\mathbb{R}$ equipped with the Euclidean distance metric [1].

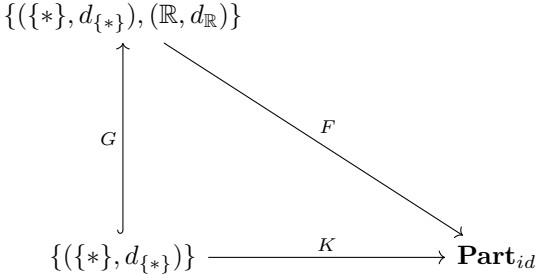

---

[1]This counterexample due to Sam Staton

Since $\mathbf{D}$ is a discrete preorder, the behavior of $K$ on $(\{*\}, d_{\{*\}})$ will not affect the behavior of the left and right Kan extensions of $K$ along $G$ on $(\mathbb{R}, d_{\mathbb{R}})$. For example, the left Kan extension of $K$ along $G$ will map $(\mathbb{R}, d_{\mathbb{R}})$ to the empty set and is therefore not well-behaved.

In order to solve this problem with Kan extensions we need to add a bit more structure. Suppose $Ob(\mathbf{D})$ is the discrete preorder with the same objects as $\mathbf{D}$ and define the following:

**Definition 4.6.** *The monotonic map $K_L : Ob(\mathbf{D}) \to \mathbf{Part}_{id}$ is equal to $K$ on $\mathbf{T}$ and maps each object $(X, d_X)$ in $Ob(\mathbf{D}) - Ob(\mathbf{T})$ to $(X, \{\{x\} \mid x \in X\})$.*

**Definition 4.7.** *The monotonic map $K_R : Ob(\mathbf{D}) \to \mathbf{Part}_{id}$ is equal to $K$ on $\mathbf{T}$ and maps each object $(X, d_X)$ in $Ob(\mathbf{D}) - Ob(\mathbf{T})$ to $(X, \{X\})$.*

Intuitively, $K_L$ and $K_R$ are extensions of $K$ to all of the objects in $\mathbf{D}$. For any metric space $(X, d_X) \in \mathbf{Met}_{id}$ not in $Ob(\mathbf{T})$ the monotonic map $K_L$ maps $(X, d_X)$ to the finest possible partition of $X$ and $K_R$ maps $(X, d_X)$ to the coarsest possible partition of $X$.

Suppose we go back to the previous example in which $\mathbf{T}$ is the discrete preorder containing only the single-element metric space and $\mathbf{D}$ is the discrete preorder containing both the single-element metric space and $(\mathbb{R}, d_{\mathbb{R}})$. Since:

$$K_L(\mathbb{R}, d_{\mathbb{R}}) = (\mathbb{R}, \{\{x\} \mid x \in \mathbb{R}\})$$

the left Kan extension of $K_L$ along the inclusion $G : Ob(\mathbf{D}) \hookrightarrow \mathbf{D}$ must map $(\mathbb{R}, d_{\mathbb{R}})$ to the $\leq_{\mathbf{Part}_{id}}$-smallest $(X, \mathbb{P}_X)$ such that:

$$(\mathbb{R}, \{\{x\} \mid x \in \mathbb{R}\}) \leq_{\mathbf{Part}_{id}} (X, \mathbb{P}_X)$$

which is $(X, \mathbb{P}_X) = (\mathbb{R}, \{\{x\} \mid x \in \mathbb{R}\})$. Similarly, since:

$$K_R(\mathbb{R}, d_{\mathbb{R}}) = (\mathbb{R}, \{\mathbb{R}\})$$

the right Kan extension of $K_R$ along the inclusion $G : Ob(\mathbf{D}) \hookrightarrow \mathbf{D}$ must map $(\mathbb{R}, d_{\mathbb{R}})$ to the $\leq_{\mathbf{Part}_{id}}$-largest $(X, \mathbb{P}_X)$ such that:

$$(X, \mathbb{P}_X) \leq_{\mathbf{Part}_{id}} (\mathbb{R}, \{\mathbb{R}\})$$

which is $(X, \mathbb{P}_X) = (\mathbb{R}, \{\mathbb{R}\})$. We can apply the same logic to the behavior of the Kan extensions on the single-element metric space as well, so both Kan extensions are well-behaved monotonic maps.

We can now build on this perspective to construct optimal extensions of $K$.

**Proposition 4.8.** *Consider the monotonic map $Lan_G K_L : \mathbf{D} \to \mathbf{Part}_{id}$ that sends the metric space $(X, d_X) \in \mathbf{D}$ to the partition of $X$ defined by the transitive closure of the relation $R$ where for $x_1, x_2 \in X$ we have $x_1 \, R \, x_2$ if and only if there exists some metric space $(X', d_{X'}) \in \mathbf{T}$ where $(X', d_{X'}) \leq_{\mathbf{D}} (X, d_X)$ and $x_1, x_2$ are in the same cluster in $K(X', d_{X'})$.*

*The map $Lan_G K_L : \mathbf{D} \to \mathbf{Part}$ is a well-behaved monotonic map. (Proof in Appendix A.1.2)*

**Proposition 4.9.** *Consider the map $Ran_G K_R : \mathbf{D} \to \mathbf{Part}_{id}$ that sends the metric space $(X, d_X) \in \mathbf{D}$ to the partition of $X$ defined by the transitive closure of the relation $R$ where for $x_1, x_2 \in X$ we have $x_1 \, R \, x_2$ if and only if there exists no metric space $(X', d_{X'}) \in \mathbf{T}$ where $(X, d_X) \leq_{\mathbf{D}} (X', d_{X'})$ and $x_1, x_2$ are in different clusters in $K(X', d_{X'})$.*

*The map $Ran_G K_R : \mathbf{D} \to \mathbf{Part}$ is a well-behaved monotonic map (Proof in Appendix A.1.3)*

We can now construct $Lan_G K_L, Ran_G K_R$ as Kan extensions.

**Proposition 4.10.** *Suppose there exists some well-behaved monotonic map $F : \mathbf{D} \to \mathbf{Part}_{id}$ where $F \circ G = K$.*

*Then $Lan_G K_L : \mathbf{D} \to \mathbf{Part}_{id}$ (Proposition 4.8) is the left Kan extension of $K_L : Ob(\mathbf{D}) \to \mathbf{Part}_{id}$ along the inclusion map $G : Ob(\mathbf{D}) \hookrightarrow \mathbf{D}$.*

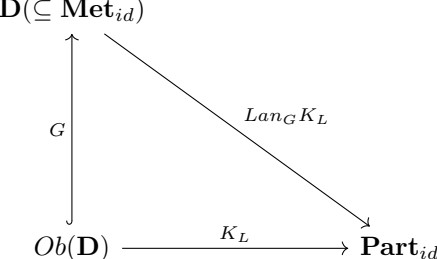

In addition $Ran_G K_R : \mathbf{D} \to \mathbf{Part}_{id}$ *(Proposition 4.9) is the right Kan extension of $K_R : Ob(\mathbf{D}) \to \mathbf{Part}_{id}$ along the inclusion map $G : Ob(\mathbf{D}) \hookrightarrow \mathbf{D}$.*

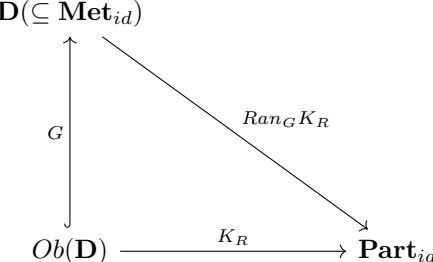

*(Proof in Appendix A.1.4)*

We will call $Lan_G K_L$ the left Kan supervised clustering map and $Ran_G K_R$ the right Kan supervised clustering map.

When $Ob(\mathbf{T}) = \emptyset$ we have for any $(X, d_X) \in \mathbf{D}$ that:

$$Lan_G K_L(X, d_X) = K_L(X, d_X) = (X, \{\{x\} \mid x \in X\})$$
$$Ran_G K_R(X, d_X) = K_R(X, d_X) = (X, \{\{X\}\})$$

In general for any metric space $(X, d_X) \in \mathbf{D} - Ob(\mathbf{T})$ the monotonic maps $Lan_G K_L, Ran_G K_R$ respectively map $(X, d_X)$ to the finest (most clusters) and coarsest (fewest clusters) partitions of $X$ such that for any metric space $(X', d_{X'}) \in \mathbf{T}$ we have:

$$K(X', d_{X'}) = Lan_G K_L(X', d_{X'}) = Ran_G K_R(X', d_{X'})$$

and $Lan_G K_L, Ran_G K_R$ are monotonic maps. For example, suppose we have a metric space $(X, d_X)$ where $X = \{x_1, x_2, x_3\}$. We can form the subpreorders $\mathbf{T} \subseteq \mathbf{D} \subseteq \mathbf{Met}_{id}$ where:

$$Ob(\mathbf{T}) = \{(\{x_1, x_2\}, d_X), (\{x_1, x_3\}, d_X), (\{x_2, x_3\}, d_X)\}$$
$$Ob(\mathbf{D}) = Ob(\mathbf{T}) \cup (\{x_1, x_2, x_3\}, d_X)$$

$\mathbf{T}$ is a discrete preorder and we define $\leq_{\mathbf{D}}$ such that $S_1 \leq_{\mathbf{D}} S_2$ when $S_1 \leq_{\mathbf{Met}_{id}} S_2$. Now define $K : \mathbf{T} \to \mathbf{Part}_{id}$ to be the following monotonic map:

$$K(\{x_1, x_2\}, d_X) = \{\{x_1, x_2\}\}$$
$$K(\{x_1, x_3\}, d_X) = \{\{x_1\}, \{x_3\}\}$$
$$K(\{x_2, x_3\}, d_X) = \{\{x_2\}, \{x_3\}\}$$

In this case we have that:

$$K_L(\{x_1, x_2, x_3\}, d_X) = \{\{x_1\}, \{x_2\}, \{x_3\}\} \qquad K_R(\{x_1, x_2, x_3\}, d_X) = \{\{x_1, x_2, x_3\}\}$$

Since the only points that need to be put together are $x_1, x_2$ and there is no metric space $(X, d_X)$ in $\mathbf{D}$ where $(\{x_1, x_2, x_3\}, d_X) <_{\mathbf{D}} (X, d_X)$ in $\mathbf{D}$, we have:

$$Lan_G K_L(\{x_1, x_2, x_3\}, d_X) = \{\{x_1, x_2\}, \{x_3\}\}$$
$$Ran_G K_R(\{x_1, x_2, x_3\}, d_X) = \{\{x_1, x_2, x_3\}\}$$

As another example, suppose $\mathbf{D}$ is $\mathbf{Met}_{id}$ and $\mathbf{T}$ is the discrete subpreorder of $\mathbf{D}$ whose objects are all metric spaces with no more than 2 elements. Define the following well-behaved monotonic map:

$$K(\{x_1, x_2\}, d) = \begin{cases} \{\{x_1, x_2\}\} & d(x_1, x_2) \leq \delta \\ \{\{x_1\}, \{x_2\}\} & \text{else} \end{cases}$$

Now for some metric space $(X, d_X) \in \mathbf{D}$ with $|X| > 2$ and points $x_1, x_2 \in X$ we have that $Lan_G K_L$ maps $x_1, x_2$ to the same cluster if and only if there exists some chain of points $x_1, \cdots, x_2$ in $X$ where for each pair of adjacent points $x_1', x_2'$ in this chain there exists some metric space $(\{x_1', x_2'\}, d_{X'}) \in \mathbf{D}$ where $(\{x_1', x_2'\}, d_{X'}) \leq_{\mathbf{D}} (X, d_X)$ and $x_1', x_2'$ are in the same cluster in $K(\{x_1', x_2'\}, d_{X'})$. This is the case if and only if $d_X(x_1', x_2') \leq \delta$. Therefore, $Lan_G K_L$ maps $x_1, x_2$ to the same cluster if and only if $x_1, x_2$ are in the same connected component of the $\delta$-Vietoris Rips complex of $(X, d_X)$. That is, $Lan_G K_L$ performs the single linkage clustering algorithm.

In contrast, since $|X| > 2$ there is no $(X', d_{X'})$ in $\mathbf{T}$ where $(X, d_X) \leq_{\mathbf{D}} (X', d_{X'})$. Therefore:

$$Ran_G K_R(X, d_X) = (X, \{X\})$$

We can use this strategy to learn a clustering algorithm from real-world data. Recall that the Fashion MNIST dataset (Xiao et al., 2017) contains images of clothing and the categories that each image falls into. Suppose that we have two subsets of this dataset: a training set $X_{tr}$ in which images are grouped by category and a testing set $X_{te}$ of ungrouped images. We can use UMAP (McInnes et al., 2018) to construct metric spaces $(X_{tr}, d_{X_{tr}})$ and $(X_{te}, d_{X_{te}})$ from these sets.

Now suppose we would like to group the images in $X_{te}$ as similarly as possible to the grouping of the images in $X_{tr}$.

For any collection of nonexpansive maps between $(X_{tr}, d_{X_{tr}})$ and $(X_{te}, d_{X_{te}})$ we can define subpreorders $\mathbf{T} \subseteq \mathbf{D} \subseteq \mathbf{Met}_{id}$ and monotonic map $K : \mathbf{T} \to \mathbf{Part}_{id}$ as follows:

1. Initialize $\mathbf{T}$ to an empty preorder and $\mathbf{D}$ to be the discrete preorder with a single object $\{(X_{te}, d_{X_{te}})\}$.

2. For every nonexpansive map $f : (X_{tr}, d_{X_{tr}}) \to (X_{te}, d_{X_{te}})$ in our collection and pair $(x_1, x_2) \in X_{tr}$ of samples in the same clothing category, add the object $(\{f(x_1), f(x_2)\}, d_{X_{te}})$ to $\mathbf{T}$ and $\mathbf{D}$ where:

$$(\{f(x_1), f(x_2)\}, d_{X_{te}}) \leq_{\mathbf{D}} (X_{te}, d_{X_{te}})$$

   and define $K(\{f(x_1), f(x_2)\}, d_{X_{te}})$ to map $f(x_1)$ and $f(x_2)$ to the same cluster.

3. For every nonexpansive map $f : (X_{te}, d_{X_{te}}) \to (X_{tr}, d_{X_{tr}})$ in our collection define a metric space $(X_{te}', d_{X_{te}'})$ where $X_{te} = X_{te}'$ and $d_{X_{te}} = d_{X_{te}'}$. Add the object $(X_{te}', d_{X_{te}'})$ to $\mathbf{T}$ and $\mathbf{D}$ where: $(X_{te}, d_{X_{te}}) \leq_{\mathbf{D}} (X_{te}', d_{X_{te}'})$ and define $K(X_{te}', d_{X_{te}'})$ to be the partition of $X_{te}'$ defined by the preimages of the function $(h \circ f)$ where $h$ maps each element of $X_{tr}$ to the category of clothing it belongs to.

We can now use $Lan_G K_L$ and $Ran_G K_R$ to partition $X_{te}$.

In Figure 1 we compare the clusterings produced by $Lan_G K_L, Ran_G K_R$ to the ground truth clothing categories. As a baseline we compute the $\delta$-single linkage clustering algorithm with $\delta$ chosen via line search to maximize the adjusted Rand score (Hubert & Arabie, 1985; Pedregosa et al., 2011) with the ground truth labels.

As expected, we see that $Lan_G K_L$ produces a finer clustering (more clusters) than does $Ran_G K_R$ and that the clusterings produced by $Lan_G K_L$ and $Ran_G K_R$ are better than the clustering produced by single linkage in the sense of adjusted Rand score with ground truth.

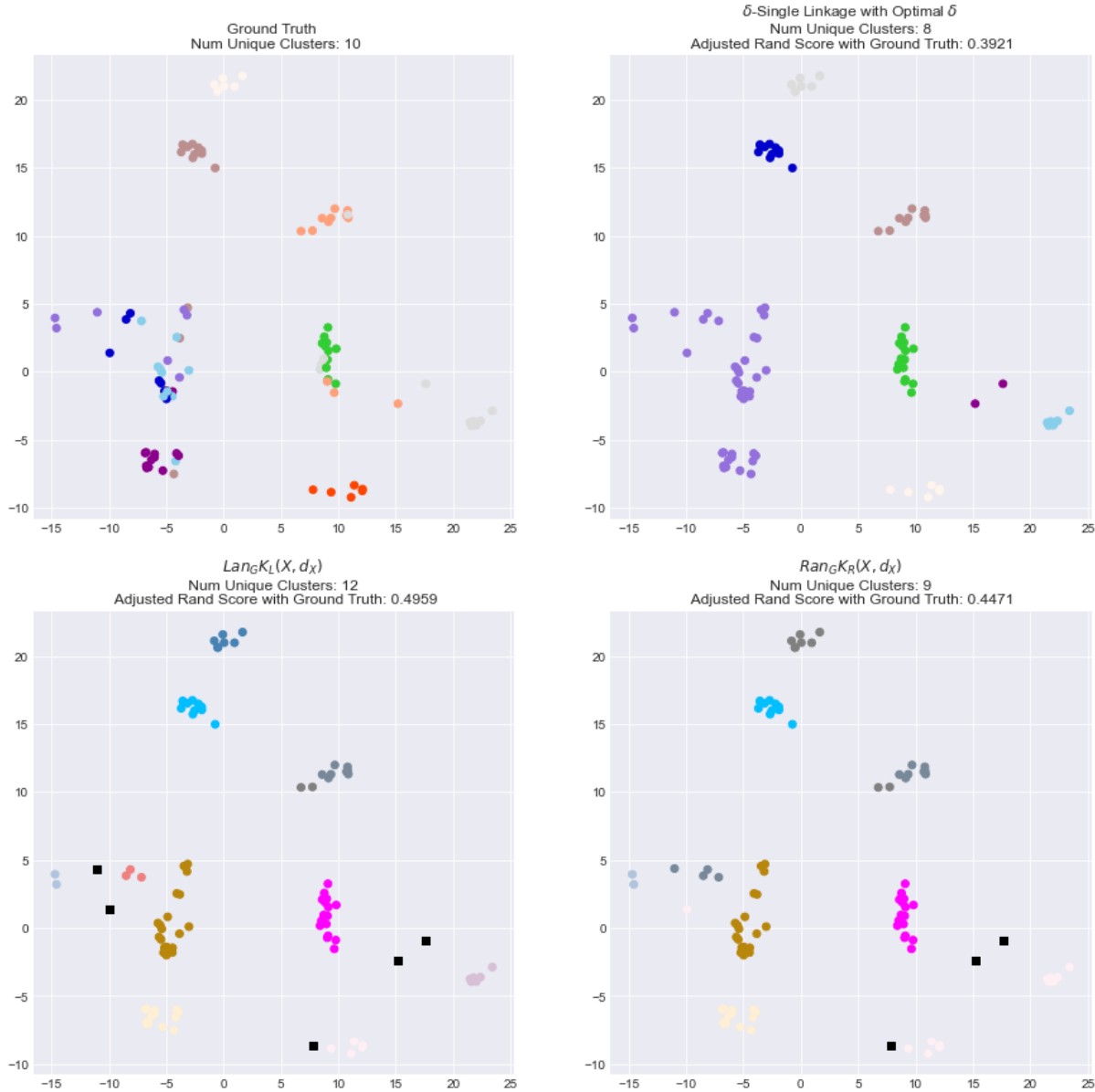

**Figure 1:** *Cluster assignments of a* 100 *point testing set* $X_{te}$ *from the Fashion MNIST dataset (Xiao et al., 2017) shown in UMAP space (McInnes et al., 2018). Each color corresponds to a unique cluster, and points without clusters are shown as black squares. We show ground truth clothing categories, unsupervised δ-single linkage cluster assignments (δ chosen via line search), and the* $Lan_G K_L, Ran_G K_R$ *supervised cluster assignments. The* $Lan_G K_L, Ran_G K_R$ *algorithms are trained on a separate* 1000 *point random sample* $X_{tr}$ *from the Fashion MNIST dataset.*

| Frequency that Left Kan Clustering Beats Best $\delta$-Single Linkage | Frequency that Right Kan Clustering Beats Best $\delta$-Single Linkage |
|:---:|:---:|
| 0.860 ($\pm$0.068) | 0.680 ($\pm$0.091) |

***Table 2:*** *We compare the performance of the left and right Kan supervised clustering maps on the Fashion MNIST dataset to the performance of $\delta$-single linkage clustering with an optimal choice of $\delta$. We select 100 bootstrap training and testing samples of the Fashion MNIST dataset. We then train and evaluate each method on each such sample. To perform this evaluation we use the scikit-learn implementation of the Adjusted Rand Score (Pedregosa et al., 2011) to compare the algorithmically generated clusterings with the ground truth categorization. We then compute the frequency with which the left and right Kan supervised clustering maps perform better (have a higher Adjusted Rand Score with ground truth) than choosing the optimal value of $\delta$ for single linkage. The win rates and two standard error confidence bounds from the 100 experiments are shown. We see that the left and right Kan clustering maps both perform consistently better than single linkage.*

## 5 Related Work

Some authors have begun to explore Kan extension structure in topological data analysis. For example, Bubenik et al. (2017) describe how three mechanisms for interpolating between persistence modules can be characterized as the left Kan extension, right Kan extension, and natural map from left to right Kan extension. Similarly, McCleary & Patel (2021) use Kan extensions to characterize deformations of filtrations. Furthermore, Botnan & Lesnick (2018) use Kan extensions to generalize stability results from block decomposable persistence modules to zigzag persistence modules and Curry (2013) uses Kan extensions to characterize persistent structures from the perspective of sheaf theory.

Other authors have explored the application of Kan extensions to databases. For example, in categorical formulations of relational database theory (Spivak & Wisnesky, 2015; Schultz & Wisnesky, 2017; Schultz et al., 2016), the left Kan extension can be used for data migration. Spivak & Wisnesky (2020) exploit the characterization of data migrations as Kan extensions to apply the chase algorithm from relational database theory to the general computation of the left Kan extension.

## 6 Future Work

In this paper we demonstrate that Kan extensions can be used to derive many different kinds of supervised learning algorithms. However, these algorithms are inherently focused on extreme values (minimums and maximums) rather than averages. Averages are required to build algorithms that are robust to noise, and a potential future direction for this work is to extend these algorithms to incorporate averages. For example, we may be able to combine multiple Kan classifiers together to generate a robust Kan classifier ensemble. It may even be possible to apply a boosting approach in which we minimize the ordering loss, fit Kan classifiers, and then repeat on the samples in the disagreement region.

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

# A  Appendix

## A.1  Proofs

### A.1.1  Proof of Proposition 3.1

*Proof.* We first need to show that $Lan_G K, Ran_G K$ are monotonic. For any $x_1 \leq x_2 \in \mathbf{I}$ suppose that $Lan_G K(x_1) = \text{true}$. Then $\exists x' \in \mathbf{I'}, x' \leq x_1, K(x') = \text{true}$. By transitivity we have $x' \leq x_2$, so:

$$Lan_G K(x_2) = \left( \begin{cases} \text{true} & \exists x' \in \mathbf{I'}, x' \leq x_2, K(x') = \text{true} \\ \text{false} & \text{else} \end{cases} \right) = \text{true}$$

and $Lan_G K$ is therefore monotonic.

Next, for any $x_1 \leq x_2 \in \mathbf{I}$ suppose that $Ran_G K(x_2) =$ false. Then $\exists x' \in \mathbf{I}', x_2 \leq x', K(x') =$ false. By transitivity we have $x_1 \leq x'$, so:

$$Ran_G K(x_1) = \left( \begin{cases} \text{false} & \exists x' \in \mathbf{I}', x_1 \leq x', K(x') = \text{false} \\ \text{true} & \text{else} \end{cases} \right) = \text{false}$$

and $Ran_G K$ is therefore monotonic.

Next we will show that $Lan_G K$ is the left Kan extension of $K$ along $G$. If for some $x' \in \mathbf{I}'$ we have that $K(x') =$ true then:

$$Lan_G K(x') = \left( \begin{cases} \text{true} & \exists x'' \in \mathbf{I}', x'' \leq x', K(x'') = \text{true} \\ \text{false} & \text{else} \end{cases} \right) = \text{true}$$

so we can conclude that $K \leq (Lan_G K \circ G)$. Now consider any other monotonic map $M_L : \mathbf{I} \to \{\text{false}, \text{true}\}$ such that $\forall x' \in \mathbf{I}', K(x') \leq M_L(x')$. We must show that $\forall x \in \mathbf{I}, Lan_G K(x) \leq M_L(x)$. For some $x \in \mathbf{I}$ suppose $M_L(x) =$ false. Then since $M_L$ is a monotonic map it must be that $\forall x' \in \mathbf{I}', x' \leq x, M_L(x') =$ false. Since $K \leq (M_L \circ G)$ it must be that $\forall x' \in \mathbf{I}', x' \leq x, K(x') =$ false. Therefore $Lan_G K(x) =$ false.

Next we will show that $Ran_G K$ is the right Kan extension of $K$ along $G$. If for some $x' \in \mathbf{I}'$ we have that $K(x') =$ false then:

$$Ran_G K(x') = \left( \begin{cases} \text{false} & \exists x'' \in \mathbf{I}', x' \leq x'', K(x'') = \text{false} \\ \text{true} & \text{else} \end{cases} \right) = \text{false}$$

so we can conclude that $(Ran_G K \circ G) \leq K$. Now consider any other monotonic map $M_R : \mathbf{I} \to \{\text{false}, \text{true}\}$ such that $\forall x' \in \mathbf{I}', M_R(x') \leq K(x')$. We must show that $\forall x \in \mathbf{I}, M_R(x) \leq Ran_G K(x)$. For some $x \in \mathbf{I}$ suppose $M_R(x) =$ true. Then since $M_R$ is a monotonic map it must be that $\forall x' \in \mathbf{I}', x \leq x', M_R(x') =$ true. Since $(M_R \circ G) \leq K$ it must be that $\forall x' \in \mathbf{I}', x \leq x', K(x') =$ true. Therefore $Ran_G K(x) =$ true. $\qquad\square$

### A.1.2   Proof of Proposition 4.8

*Proof.* $Lan_G K_L$ trivially acts as the identity on underlying sets so we simply need to show that when:

$$(X, d_X) \leq_{\mathbf{D}} (Y, d_Y)$$

then

$$Lan_G K_L(X, d_X) \leq_{\mathbf{Part}_{id}} Lan_G K_L(Y, d_Y)$$

Suppose there exists some $x, x^* \in X$ in the same cluster in $Lan_G K_L(X, d_X)$. Then by the definition of $Lan_G K_L$ there must exist some sequence

$$(X_1, d_{X_1}), (X_2, d_{X_2}), \cdots, (X_n, d_{X_n}) \in \mathbf{T}$$

where $x \in X_1, x^* \in X_n$ and each:

$$(X_i, d_{X_i}) \leq_{\mathbf{D}} (X, d_X)$$

as well as some sequence

$$x_1, x_2, \cdots, x_{n-1}, \text{ such that } x_i \in X_i, x_i \in X_{i+1}$$

where the pair $(x, x_1)$ is in the same cluster in $K(X_1, d_{X_1})$, the pair $(x_{n-1}, x^*)$ is in the same cluster in $K(X_n, d_{X_n})$, and for each $1 < i < n$ the pair $(x_{i-1}, x_i)$ is in the same cluster in $K(X_i, d_{X_i})$. Since it must be that each:

$$(X_i, d_{X_i}) \leq_{\mathbf{D}} (Y, d_Y)$$

as well then by the definition of $Lan_G K_L$ it must be that $x, x^*$ are in the same cluster in $Lan_G K_L(Y, d_Y)$. $\quad\square$

### A.1.3 Proof of Proposition 4.9

*Proof.* $Ran_G K_R$ trivially acts as the identity on underlying sets so we simply need to show that when:

$$(X, d_X) \leq_\mathbf{D} (Y, d_Y)$$

then:

$$Ran_G K_R(X, d_X) \leq_{\mathbf{Part}_{id}} Ran_G K_R(Y, d_Y)$$

Suppose the points $x, x^* \in X$ are in the same cluster in $Ran_G K_R(X, d_X)$. Then by the definition of $Ran_G K_R$ there cannot be any $(X', d_{X'})$ in $\mathbf{T}$ such that:

$$(X, d_X) \leq_\mathbf{D} (X', d_{X'})$$

and $x, x^*$ are in different clusters in $Ran_G K_R(X', d_{X'})$. By transitivity this implies that there cannot be any $(X'', d_{X''})$ in $\mathbf{T}$ such that:

$$(Y, d_Y) \leq_\mathbf{D} (X'', d_{X''})$$

and $x, x^*$ are in different clusters in $Ran_G K_R(X'', d_{X''})$. By the definition of $Ran_G K_R$ the points $x, x^*$ must therefore be in the same cluster in $Ran_G K_R(Y, d_Y)$. $\qquad\square$

### A.1.4 Proof of Proposition 4.10

We use the following Proposition in the proof below:

**Proposition A.1.** *Suppose there exists some well-behaved monotonic map $F : \mathbf{D} \to \mathbf{Part}_{id}$ where $F \circ G = K$. Then for $(X, d_X) \in \mathbf{T}$ we have that:*

$$F(X, d_X) = K(X, d_X) = Lan_G K_L(X, d_X) = Ran_G K_R(X, d_X)$$

*Proof.* Since each of:

$$F : \mathbf{D} \to \mathbf{Part}$$
$$Ran_G K_R : \mathbf{D} \to \mathbf{Part}$$
$$Lan_G K_L : \mathbf{D} \to \mathbf{Part}$$

are well-behaved monotonic maps we simply need to prove that all three maps generate the same partition of $X$ for any input $(X, d_X) \in \mathbf{T}$.

Consider some $(X, d_X) \in \mathbf{T}$ and two points $x, x^* \in X$. Suppose $x, x^*$ are in different clusters in

$$K(X, d_X) = F(X, d_X)$$

. Then since $F$ is a well-behaved monotonic map it must be that for any sequence

$$(X_1, d_{X_1}), (X_2, d_{X_2}), \cdots, (X_n, d_{X_n}) \in \mathbf{T}$$

where $x \in X_1, x^* \in X_n$ and each:

$$(X_i, d_{X_i}) \leq_\mathbf{D} (X, d_X)$$

and any sequence

$$x_1, x_2, \cdots, x_{n-1}, \text{ such that } x_i \in X_i, x_i \in X_{i+1}$$

one of the following must be true:

- The pair $(x, x_1)$ are in different clusters in $F(X_1, d_{X_1})$

- The pair $(x_{n-1}, x^*)$ are in different clusters in $F(X_n, d_{X_n})$

- For some $1 < i < n$ the pair $(x_{i-1}, x_i)$ are in different clusters in $F(X_i, d_{X_i})$

This implies that in $Lan_G K_L(X, d_X)$ the points $x, x^*$ must be in different clusters. Similarly, since $(X, d_X) \leq_{\mathbf{D}} (X, d_X)$, by Proposition 4.9 it must be that $x, x^*$ are in different clusters in $Ran_G K_R(X, d_X)$.

Now suppose $x, x^*$ are in the same cluster in:

$$K(X, d_X) = F(X, d_X)$$

Since $(X, d_X) \leq_{\mathbf{D}} (X, d_X)$, by Proposition 4.8 it must be that $x, x^*$ are in the same cluster in $Lan_G K_L(X, d_X)$. Similarly, since $F$ is a well-behaved monotonic map there cannot exist any metric space $(X', d_{X'}) \in \mathbf{T}$ where:

$$(X, d_X) \leq_{\mathbf{D}} (X', d_{X'})$$

and $x, x^*$ are in different clusters in:

$$K(X', d_{X'}) = F(X', d_{X'})$$

Therefore $x, x^*$ are in the same cluster in $Ran_G K_R(X, d_X)$. □

Now we can prove Proposition 4.10:

*Proof.* To start, note that Proposition A.1 implies that for any $(X, d_X) \in \mathbf{T}$ we have:

$$Lan_G K_L(X, d_X) = K(X, d_X) = Ran_G K_R(X, d_X)$$

By the definition of $K_L, K_R$ we can therefore conclude that for any $(X, d_X) \in \mathbf{D}$ we have:

$$K_L(X, d_X) \leq_{\mathbf{Part}_{id}} Lan_G K_L(X, d_X)$$
$$Ran_G K_R(X, d_X) \leq_{\mathbf{Part}_{id}} K_R(X, d_X)$$

Next, consider any monotonic map $M_L : \mathbf{D} \to \mathbf{Part}_{id}$ such that for all $(X, d_X) \in \mathbf{D}$ we have:

$$K_L(X, d_X) \leq_{\mathbf{Part}_{id}} (M_L \circ G)(X, d_X)$$

We must show that for any $(X, d_X) \in \mathbf{D}$ we have:

$$Lan_G K_L(X, d_X) \leq_{\mathbf{Part}_{id}} M_L(X, d_X)$$

To start, note that for any $x, x^* \in X$ that are in the same cluster in $Lan_G K_L(X, d_X)$ by the definition of $Lan_G K_L$ there must exist some sequence:

$$(X_1, d_{X_1}), (X_2, d_{X_2}), \cdots, (X_n, d_{X_n}) \in \mathbf{T}$$

where $x \in X_1, x^* \in X_n$ and each:

$$(X_i, d_{X_i}) \leq_{\mathbf{D}} (X, d_X)$$

as well as some sequence

$$x_1, x_2, \cdots, x_{n-1}, \text{ such that } x_i \in X_i, x_i \in X_{i+1}$$

where the pair $(x, x_1)$ is in the same cluster in $K_L(X_1, d_{X_1})$, the pair $(x_{n-1}, x^*)$ is in the same cluster in $K_L(X_n, d_{X_n})$, and for each $1 < i < n$ the pair $(x_{i-1}, x_i)$ is in the same cluster in $K_L(X_i, d_{X_i})$. Now since for each $(X_i, d_{X_i})$ in this sequence we have that:

$$K_L(X_i, d_{X_i}) \leq_{\mathbf{Part}_{id}} M_L(X_i, d_{X_i})$$

it must be that the pair $(x, x_1)$ is in the same cluster in $M_L(X_1, d_{X_1})$, the pair $(x_{n-1}, x^*)$ is in the same cluster in $M_L(X_n, d_{X_n})$, and for each $1 < i < n$ the pair $(x_{i-1}, x_i)$ is in the same cluster in $M_L(X_i, d_{X_i})$.

Since $M_L$ is a monotonic map it must therefore be that the pair $x, x^*$ is in the same cluster in $M_L(X, d_X)$ and therefore:

$$Lan_G K_L(X, d_X) \leq_{\mathbf{Part}_{id}} M_L(X, d_X)$$

Next, consider any monotonic map $M_R : \mathbf{D} \to \mathbf{Part}_{id}$ such that for all $(X, d_X)$ in $\mathbf{D}$:

$$(M_R \circ G)(X, d_X) \leq_{\mathbf{Part}_{id}} K_R(X, d_X)$$

We must show that for any $(X, d_X)$ in $\mathbf{D}$ we have:

$$M_R(X, d_X) \leq_{\mathbf{Part}_{id}} Ran_G K_R(X, d_X)$$

To start, note that for any $x, x^* \in X$ such that $x, x^*$ are not in the same cluster in $Ran_G K_R(X, d_X)$ by the definition of $Ran_G K_R$ there must exist some:

$$(X', d_{X'}) \in \mathbf{D}, (X, d_X) \leq_{\mathbf{D}} (X', d_{X'})$$

where $x, x^*$ are not in the same cluster in $K_R(X', d_{X'})$. Now since:

$$M_R(X', d_{X'}) \leq_{\mathbf{Part}_{id}} K_R(X', d_{X'})$$

it must be that $x, x^*$ are not in the same cluster in $M_R(X', d_{X'})$. Since $M_R$ is a monotonic map we have:

$$M_R(X, d_X) \leq_{\mathbf{Part}_{id}} M_R(X', d_{X'})$$

so $x, x^*$ are also not in the same cluster in $M_R(X, d_X)$ and therefore:

$$M_R(X, d_X) \leq_{\mathbf{Part}_{id}} Ran_G K_R(X, d_X)$$

$\square$

## A.2 Meta-Supervised Learning

Suppose $I$ is a set and $O$ is a partial order. A supervised learning algorithm maps a labeled dataset (set of pairs of points in $I \times O$) to a function $f : I \to O$. For example, both $Lan_G K$ and $Ran_G K$ from Section 3 are supervised learning algorithms.

In this section we use Kan extensions to derive supervised learning algorithms from pairs of datasets and functions. Our construction combines elements of Section 3's point-level algorithms and Section 4's dataset-level algorithms.

Suppose we have a finite partial order $S_f \subseteq (I \to O)$ of functions where for $f, f' \in S_f$ we have $f \leq f'$ when $\forall x \in I, f(x) \leq f'(x)$.

**Proposition A.2.** *For any subset $S_f^* \subseteq S_f$ the upper antichain of $S_f^*$ is the set:*

$$\{f \mid f \in S_f^*, \ \nexists f^* \in S_f^*, f < f^*\}\}$$

*The upper antichain of $S_f^*$ is an antichain in $S_f^*$, and for any function $f \in S_f^*$ there exists some function $f^*$ in the upper antichain of $S_f^*$ such that $f \leq f^*$.*

*Proof.* Suppose $f_1, f_2$ are in the upper antichain of $S_f^* \subseteq S_f$ and $f_1 \leq f_2$. Then since

$$\nexists f_1^* \in S_f^*, f_1 < f_1^*$$

it must be that $f_1 = f_2$ and we can conclude that the upper antichain is an antichain.

Next, for any function $f \in S_f^*$ consider the set $\{f^* \in S_f^*, f < f^*\}$. Since $S_f$ is finite this set must have finite size. If this set is empty then $f$ is in the upper antichain of $S_f^*$. If this set has size $n$ then for any $f^*$ in this set the set $\{f^{**} \in S_f^*, f^* < f^{**}\}$ must have size strictly smaller than $n$. We can therefore conclude by induction that the upper antichain of $S_f^*$ contains at least one function $f^*$ where $f \leq f^*$. $\qquad \square$

Intuitively the upper antichain of $S_f^*$ is the collection of all functions $f \in S_f^*$ that are not strictly upper bounded by any other function in $S_f^*$. The upper antichain of an empty set is of course itself an empty set.

**Definition A.3.** *We can form the following preorders:*

$\mathbf{D_C}$ *: The objects in $\mathbf{D_C}$ are $\leq$-antichains of functions $X_f \subseteq S_f$. $\mathbf{D_C}$ is a preorder in which $X_f \leq X_f'$ if for $f \in X_f$ there must exist some $f' \in X_f'$ where $f \leq f'$.*

$\mathbf{D_B}$ *: The objects in $\mathbf{D_B}$ are labeled datasets, or sets of pairs $U = \{(x, y) \mid x \in I, y \in O\}$. $\mathbf{D_B}$ is a preorder such that $U \leq U'$ when for all $(x, y') \in U'$ there exists $(x, y) \in U$ where $y \leq y'$.*

$\mathbf{D_A}$ *: A subpreorder of $\mathbf{D_B}$ such that if $U \leq U' \in \mathbf{D_B}$ then $U \leq U' \in \mathbf{D_A}$.*

**Proposition A.4.** $\mathbf{D_B}$ *and* $\mathbf{D_C}$ *are preorders.*

*Proof.*
$\mathbf{D_C}$
We trivially have $X_f \leq X_f$ in $\mathbf{D_C}$. To see that $\leq$ is transitive in $\mathbf{D_C}$ simply note that if $X_{f_1} \leq X_{f_2}$ and $X_{f_2} \leq X_{f_3}$ then for $f_1 \in X_{f_1}$ there must exist $f_2 \in X_{f_2}, f_1 \leq f_2$, which implies that there must exist $f_3 \in X_{f_3}, f_1 \leq f_2 \leq f_3$.

$\mathbf{D_B}$
We trivially have $U \leq U$ in $\mathbf{D_B}$. To see that $\leq$ is transitive in $\mathbf{D_B}$ simply note that if $U_1 \leq U_2$ and $U_2 \leq U_3$ in $\mathbf{D_B}$ then for $(x, y_3) \in U_3$ there must exist $(x, y_2) \in U_2, y_2 \leq y_3$ which implies that there must exist $(x, y_1) \in U_1, y_1 \leq y_2 \leq y_3$. $\qquad \square$

Intuitively, $\mathbf{D_A}$ is a collection of labeled training datasets and $\mathbf{D_B}$ is a collection of labeled testing datasets. We can define a monotonic map that maps each training dataset to all of the trained models that agree with that dataset.

**Proposition A.5.** *The map $K : \mathbf{D_A} \to \mathbf{D_C}$ that maps the object $U \in \mathbf{D_A}$ to the upper antichain of the following set:*

$$S_K(U) = \{f \mid f \in S_f, \forall (x, y) \in U, f(x) \leq y\}$$

*is a monotonic map.*

*Proof.* To start, note that $K$ maps objects in $\mathbf{D_A}$ to objects in $\mathbf{D_C}$ since the upper antichain of $S_K(U)$ must be an antichain in $S_f$ by Proposition A.2.

Next, we need to show that if $U \leq U'$ then $K(U) \leq K(U')$. For any $x, y' \in U'$ it must be that there exists $(x, y) \in U$ where $y \leq y'$, so if $f \in K(U)$ then by the definition of $K$ we have $f(x) \leq y \leq y'$. Therefore $f \in S_K(U')$, so by Proposition A.2 $K(U')$ contains $f'$ where $f \leq f'$. Therefore $K(U) \leq K(U')$. $\qquad \square$

Now define $G : \mathbf{D_A} \hookrightarrow \mathbf{D_B}$ to be the inclusion map. A monotonic map $F : \mathbf{D_B} \to \mathbf{D_C}$ such that $F \circ G$ commutes with $K$ will then be a mapping from the testing datasets in $\mathbf{D_B}$ to collections of trained models.

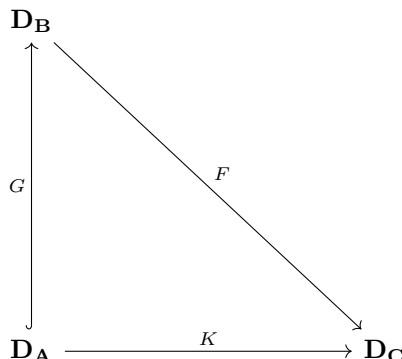

We can take the left and right Kan extensions of $K$ along the inclusion map $G : \mathbf{D_A} \hookrightarrow \mathbf{D_B}$ to find the optimal such mapping.

**Proposition A.6.** *The map $Lan_G K$ that maps the object $U \in \mathbf{D_B}$ to the upper antichain of the following set:*

$$S_L(U) = \bigcup_{\{U' \mid U' \in \mathbf{D_A}, U' \leq U\}} K(U')$$

*is the left Kan extension of $K$ along $G$.*

*Next, the map $Ran_G K$ that maps the object $U \in \mathbf{D_B}$ to the upper antichain of the following set:*

$$S_R(U) = \{f \mid f \in S_f,\ \forall U' \in \{U' \mid U' \in \mathbf{D_A}, U \leq U'\},\ \exists f' \in K(U'),\ f \leq f'\}$$

*is the right Kan extension of $K$ along $G$.*

*Proof.* We first need to show that $Lan_G K$ is a monotonic map $\mathbf{D_B} \to \mathbf{D_C}$. Note that $Lan_G K$ maps objects in $\mathbf{D_B}$ to objects in $\mathbf{D_C}$ since the upper antichain of $S_L(U)$ must be an antichain in $S_f$.

Next, suppose $U_1 \leq U_2$ and that $f \in Lan_G K(U_1)$. Consider the set of all $U' \in \mathbf{D_A}$ where $U' \leq U_1$. Since $U_1 \leq U_2$ this is a subset of the set of all $U' \in \mathbf{D_A}$ where $U' \leq U_2$. Since $S_L(U_1)$ is defined to be a union of the elements in the set we have that $S_L(U_1) \subseteq S_L(U_2)$. Since $f \in Lan_G K(U_1)$ implies that $f \in S_L(U_1)$ this implies that $f \in S_L(U_2)$ as well. Proposition A.2 then implies that there must exist $f' \in Lan_G K(U_2)$ where $f \leq f'$ and therefore $Lan_G K(U_1) \leq Lan_G K(U_2)$.

Next, we will show that $Lan_G K$ is the left Kan extension of $K$ along $G$.

- Consider some $U \in \mathbf{D_A}$ and $f \in K(U)$. Since $U \leq U$ we have by the definition of $S_L$ that $f \in S_L(U)$. Proposition A.2 then implies that $\exists f' \in Lan_G K(U)$ such that $f \leq f'$. This implies that $K \leq Lan_G K \circ G$.

- Now consider any monotonic map $M_L : \mathbf{D_B} \to \mathbf{D_C}$ such that $K \leq (M_L \circ G)$. We must show that $Lan_G K \leq M_L$. For some $U \in \mathbf{D_B}$ suppose $f \in Lan_G K(U)$. By the definition of $S_L$ there must exist some $U' \in \mathbf{D_A}$ where $U' \leq U$ such that $f \in K(U')$. Since $K(U') \leq M_L(U')$ there must exist some $f' \in M_L(U')$ where $f \leq f'$. Since $M_L$ is a monotonic map we have $M_L(U') \leq M_L(U)$ which implies that there must exist some $f^* \in M_L(U)$ where $f \leq f' \leq f^*$. Therefore $Lan_G K \leq M_L$.

Next, we need to show that $Ran_G K$ is a monotonic map $\mathbf{D_B} \to \mathbf{D_C}$. Note that $Ran_G K$ maps objects in $\mathbf{D_B}$ to objects in $\mathbf{D_C}$ since the upper antichain of $S_R(U)$ must be an antichain in $S_f$.

Next, suppose $U_1 \leq U_2$ and that $f \in Ran_G K(U_1)$. Consider the set of all $U' \in \mathbf{D_A}$ where $U_2 \leq U'$. Since $U_1 \leq U_2$ this is a subset of the set of all $U' \in \mathbf{D_A}$ where $U_1 \leq U'$. Therefore by the definition of $S_R$ we have that $S_R(U_1) \subseteq S_R(U_2)$. Since $f \in Ran_G K(U_1)$ implies that $f \in S_R(U_1)$ this implies that $f \in S_R(U_2)$ as well. Proposition A.2 then implies that there must exist $f' \in Ran_G K(U_2)$ where $f \leq f'$ and therefore $Ran_G K(U_1) \leq Ran_G K(U_2)$.

Next, we will show that $Ran_G K$ is the right Kan extension of $K$ along $G$.

- For $U \in \mathbf{D_A}$ since $U \leq U$ when $f \in S_R(U)$ we have by the definition of $S_R$ that $\exists f' \in K(U)$ such that $f \leq f'$. Since $Ran_G K(U)$ is a subset of $S_R(U)$ this implies that $Ran_G K \circ G \leq K$.

- Now consider any monotonic map $M_R : \mathbf{D_B} \to \mathbf{D_C}$ such that $(M_R \circ G) \leq K$. We must show that $M_R \leq Ran_G K$. For some $U \in \mathbf{D_B}$ suppose $f \in M_R(U)$. Since $M_R$ is a monotonic map it must be that for all $U' \in \mathbf{D_A}$ where $U \leq U'$ we have that $M_R(U) \leq M_R(U')$ and therefore $\exists f'_{M_R} \in M_R(U'), f \leq f'_{M_R}$. Since $(M_R \circ G) \leq K$ this implies that for all $U' \in \mathbf{D_A}$ where $U \leq U'$ we have that $\exists f'_K \in K(U'), f \leq f'_{M_R} \leq f'_K$. By the definition of $S_R$ this implies that $f \in S_R(U)$. Proposition A.2 therefore implies that there exists $f'_R \in Ran_G K(U)$ such that $f \leq f'_R$, and therefore $M_R(U) \leq Ran_G K(U)$.

$\square$

Intuitively the functions in $Ran_G K(U)$ and $Lan_G K(U)$ are as large as possible subject to constraints imposed by the selection of sets in $Ob(\mathbf{D_A})$. The functions in $Lan_G K(U)$ are subject to a membership constraint and grow smaller when we remove objects from $Ob(\mathbf{D_A})$. The functions in $Ran_G K(U)$ are subject to an upper boundedness-constraint and grow larger when we remove objects from $Ob(\mathbf{D_A})$.

Consider the extreme case where $Ob(\mathbf{D_A}) = \emptyset$. For any $U \in \mathbf{D_B}$ we have that:

$$S_L(U) = \bigcup_{\{U' \mid U' \in \emptyset, \cdots\}} K(U') = \emptyset$$
$$S_R(U) = \{f \mid f \in S_f, \forall U' \in \emptyset, \cdots\} = S_f$$

so $Lan_G K(U)$ is empty and $Ran_G K(U)$ is the upper antichain of $S_f$.

Now consider the extreme case where $Ob(\mathbf{D_A}) = Ob(\mathbf{D_B})$. For any $U \in \mathbf{D_B}$ and $f \in K(U)$ the monotonicity of $K$ implies that:

$$\forall U' \in \{U' \mid U' \in \mathbf{D_A}, U \leq U'\}, \exists f' \in K(U'), f \leq f'$$

and therefore $f \in S_R(U)$. This implies $K(U) \leq Ran_G K(U)$. Similarly, for any $f \in Lan_G K(U)$ it must be that:

$$\exists U' \in \mathbf{D_A}, U' \leq U, f \in K(U')$$

which by the monotonicity of $K$ implies that:

$$\exists f^* \in K(U), f \leq f^*$$

and therefore $Lan_G K(U) \leq K(U)$. Therefore in this extreme case we have:

$$Ran_G K(U) = Lan_G K(U) = K(U)$$

Let's now consider a more concrete example. Suppose $I = \mathbb{R}^2_{\geq 0}, O = \{\text{false}, \text{true}\}$, and $S_f$ is the finite set of linear classifiers $l : \mathbb{R}^2_{\geq 0} \to \{\text{false}, \text{true}\}$ that can be expressed as:

$$l_{a,b}(x_1, x_2) = \begin{cases} \text{true} & x_2 \leq a * x_1 + b \\ \text{false} & \text{else} \end{cases}$$

where $a, b$ are integers in $(-100, 100)$. Intuitively:

- The classifiers in $Lan_G K(U)$ are selected to be the classifiers that predict true as often as possible among the set of all classifiers that have no false positives on some $U' \in \mathbf{D_A}$ where $U' \leq U$.

- The classifiers in $Ran_G K(U)$ are constructed to predict true as often as possible subject to a constraint imposed by the selection of sets in $\mathbf{D_A}$. For every set $U' \in \mathbf{D_A}$ where $U \leq U'$ it must be that each classifier in $Ran_G K(U)$ is upper bounded at each point in $I$ by some classifier in $S_f$ with no false positives on $U'$.

A concrete example will demonstrate this. Suppose that $\mathbf{D_A}$ is:

$$\{((2,2), \text{true}), ((1,3), \text{true}), ((4,4), \text{false})\}$$

$$\leq \uparrow$$

$$\{((2,2), \text{true}), ((1,3), \text{false}), ((4,4), \text{true})\} \xleftarrow{\leq} \{((2,2), \text{false}), ((1,3), \text{false}), ((4,4), \text{false})\}$$

and that $\mathbf{D_B}$ is:

$$\{((2,2), \text{true}), ((1,3), \text{true}), ((4,4), \text{false})\}$$

$$\leq \uparrow$$

$$\{((2,2), \text{true}), ((1,3), \text{false}), ((4,4), \text{true})\} \xleftarrow{\leq} \{((2,2), \text{true}), ((1,3), \text{false}), ((4,4), \text{false})\}$$

$$\leq \uparrow$$

$$\{((2,2), \text{false}), ((1,3), \text{false}), ((4,4), \text{false})\}$$

We can see the following:

- $l_{(1,1)} \in K(\{((2,2), \text{true}), ((1,3), \text{false}), ((4,4), \text{true})\})$ since:

$$l_{(1,1)}(1,3) = \left( \begin{cases} \text{true} & 3 \leq 1*1+1 \\ \text{false} & \text{else} \end{cases} \right) = \text{false}$$

but we have that:

$$l_{(1,2)}(1,3) = \left( \begin{cases} \text{true} & 3 \leq 1*1+2 \\ \text{false} & \text{else} \end{cases} \right) = \text{true}$$

$$l_{(2,1)}(1,3) = \left( \begin{cases} \text{true} & 3 \leq 2*1+1 \\ \text{false} & \text{else} \end{cases} \right) = \text{true}$$

- $l_{(0,2)} \in K(\{((2,2), \text{true}), ((1,3), \text{false}), ((4,4), \text{true})\})$ since:

$$l_{(0,2)}(1,3) = \left( \begin{cases} \text{true} & 3 \leq 0*1+2 \\ \text{false} & \text{else} \end{cases} \right) = \text{false}$$

but we have that:

$$l_{(0,3)}(1,3) = \left( \begin{cases} \text{true} & 3 \leq 0*1+3 \\ \text{false} & \text{else} \end{cases} \right) = \text{true}$$

$$l_{(1,2)}(1,3) = \left( \begin{cases} \text{true} & 3 \leq 1*1+2 \\ \text{false} & \text{else} \end{cases} \right) = \text{true}$$

- $l_{(0,3)} \in K(\{((2,2), \text{true}), ((1,3), \text{true}), ((4,4), \text{false})\})$ since:

$$l_{(0,3)}(4,4) = \left( \begin{cases} \text{true} & 4 \leq 0 * 4 + 3 \\ \text{false} & \text{else} \end{cases} \right) = \text{false}$$

but we have that:

$$l_{(1,3)}(4,4) = \left( \begin{cases} \text{true} & 4 \leq 1 * 4 + 3 \\ \text{false} & \text{else} \end{cases} \right) = \text{true}$$

$$l_{(0,4)}(4,4) = \left( \begin{cases} \text{true} & 4 \leq 0 * 4 + 4 \\ \text{false} & \text{else} \end{cases} \right) = \text{true}$$

- $l_{(0,1)} \in K(\{((2,2), \text{false}), ((1,3), \text{false}), ((4,4), \text{false})\})$ since:

$$l_{(0,1)}(2,2) = \left( \begin{cases} \text{true} & 2 \leq 0 * 2 + 1 \\ \text{false} & \text{else} \end{cases} \right) = \text{false}$$

$$l_{(0,1)}(1,3) = \left( \begin{cases} \text{true} & 3 \leq 0 * 1 + 1 \\ \text{false} & \text{else} \end{cases} \right) = \text{false}$$

$$l_{(0,1)}(4,4) = \left( \begin{cases} \text{true} & 4 \leq 0 * 4 + 1 \\ \text{false} & \text{else} \end{cases} \right) = \text{false}$$

but we have that:

$$l_{(1,1)}(4,4) = \left( \begin{cases} \text{true} & 4 \leq 1 * 4 + 1 \\ \text{false} & \text{else} \end{cases} \right) = \text{true}$$

$$l_{(0,2)}(2,2) = \left( \begin{cases} \text{true} & 2 \leq 0 * 2 + 2 \\ \text{false} & \text{else} \end{cases} \right) = \text{true}$$

By the definition of $Lan_G K$ we have that:

$$Lan_G K(\{((2,2), \text{true}), ((1,3), \text{false}), ((4,4), \text{false})\})$$

must contain $l_{(0,1)}$ since we have that:

$$l_{(0,1)} \in K(\{((2,2), \text{false}), ((1,3), \text{false}), ((4,4), \text{false})\})$$

but:

$$l_{(0,2)} \notin K(\{((2,2), \text{false}), ((1,3), \text{false}), ((4,4), \text{false})\})$$
$$l_{(1,1)} \notin K(\{((2,2), \text{false}), ((1,3), \text{false}), ((4,4), \text{false})\})$$

Similarly, by the definition of $Ran_G K$ we have that:

$$Ran_G K(\{((2,2), \text{true}), ((1,3), \text{false}), ((4,4), \text{false})\})$$

must contain $l_{(0,2)}$ since we have that:

$$l_{(0,2)} \leq l_{(0,3)} \qquad l_{(0,2)} \leq l_{(1,2)}$$

but that there is no $l_{(a,b)}$ such that $l_{(0,2)} < l_{(a,b)}$ that is in both:

$$K(\{((2,2), \text{true}), ((1,3), \text{true}), ((4,4), \text{false})\})$$

and:

$$K(\{((2,2), \text{true}), ((1,3), \text{false}), ((4,4), \text{true})\})$$

since:

$$l_{(1,2)} \notin K(\{((2,2), \text{true}), ((1,3), \text{true}), ((4,4), \text{false})\})$$
$$l_{(0,3)} \notin K(\{((2,2), \text{true}), ((1,3), \text{false}), ((4,4), \text{true})\})$$

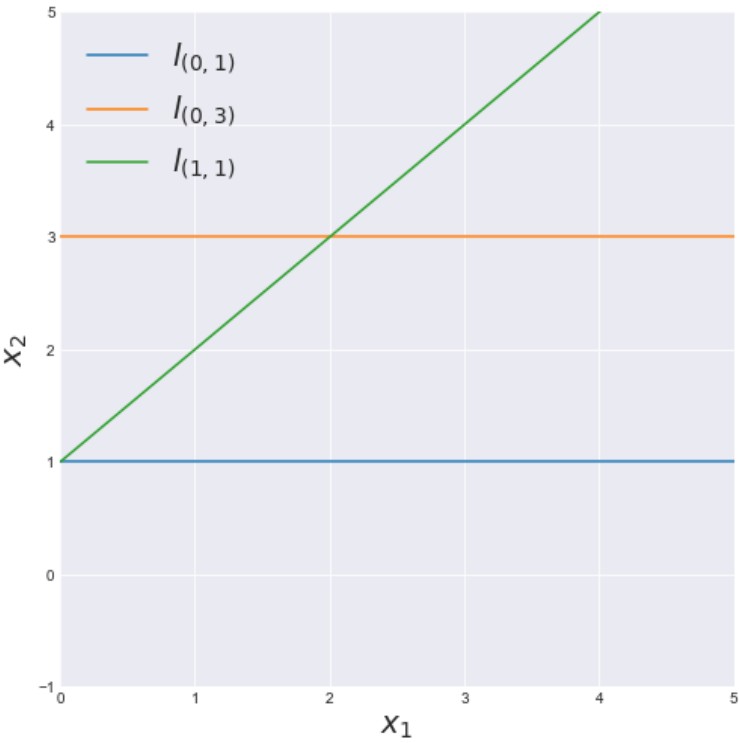

***Figure 2:*** *The decision boundaries defined by $l_{(1,1)}$, $l_{(0,3)}$, and $l_{(0,1)}$.*

### A.3 Function Approximation

In many learning applications there may be multiple functions in a class that fit a particular set of data similarly well. In such a situation Occam's Razor suggests that we are best off choosing the simplest such function. For example, we can choose the function with the smallest Kolmogorov complexity, also known as the minimum description length (MDL) function (Rissanen, 1978). In this section we will explore how we can use Kan extensions to find the MDL function that fits a dataset.

Suppose $I$ is a set, $O$ is a partial order, and $S$ is a finite subset of $I$. We can define the following preorder:

**Definition A.7.** *Define the preorder $\leq_S$ on $(I \to O)$ such that $f_1 \leq_S f_2$ if and only if $\forall x \in S, f_1(x) \leq f_2(x)$. If $f_1 \leq_S f_2, f_2 \nleq_S f_1$ then write $f_1 <_S f_2$ and if $f_1 \leq_S f_2 \leq_S f_1$ then write $f_1 =_S f_2$.*

Now suppose also that $C_{\leq_c}$ is some finite subset of the space of all functions $(I \to O)$ equipped with a total order $\leq_c$ such that $f_1 \leq_c f_2$ whenever the Kolmogorov complexity of $f_1$ is no larger than that of $f_2$. Note that functions with the same Kolmogorov complexity may be ordered arbitrarily in $C_{\leq_c}$.

**Proposition A.8.** *Given a set of functions $S_f \subseteq C_{\leq_c}$ we can define a map that sends each function $f \in S_f$ to the function:*

$$f_c = \min_{\leq_c}\{f' \mid f' \in S_f, f' =_S f\}$$

*where $f_c$ satisfies $f_c \leq_c f$.*

*This map is guaranteed to exist and we can define the minimum Kolmogorov subset $S_{f_c}$ of $S_f$ to be the image of this map. $S_{f_c}$ contains exactly one function $f_c$ where $f =_S f_c$.*

*Proof.* For any function $f \in S_f$ there must exist some $f_c = \min_{\leq_c}\{f' \mid f' \in S_f, f' =_S f\}$ since $\{f' \mid f' \in S_f, f' =_S f\}$ is a nonempty finite total $\leq_c$-order. Therefore we can define a map that sends each $f \in S_f$ to $f_c$ and we can define $S_{f_c}$ to be the image of this map.

Since this map will send all $f \in S_f$ in the same $=_S$ equivalence class to the same function in that $=_S$ equivalence class, $S_{f_c}$ contains exactly one function $f_c$ where $f =_S f_c$. This function $f_c$ satisfies $f_c \leq_c f$. $\square$

We can use these constructions to define the following preorders:

**Definition A.9.** *Given the sets of functions $S_f^1 \subseteq S_f^2 \subseteq C_{\leq_c}$ define $S_{f_c}^1$ to be the minimum Kolmogorov subset of $S_f^1$. We can construct the preorders $\mathbf{F_A}, \mathbf{F_B}, \mathbf{F_C}$ as follows.*

- *The set of objects in the discrete preorder $\mathbf{F_A}$ is $S_{f_c}^1$.*

- *The set of objects in $\mathbf{F_B}$ is $S_f^2$. $\mathbf{F_B}$ is a preorder under $\leq_S$.*

- *$\mathbf{F_C}$ is the subpreorder of $\mathbf{F_B}$ under $\leq_S$ in which objects are functions in $S_{f_c}^1$.*

Intuitively a monotonic map $\mathbf{F_B} \to \mathbf{F_C}$ acts as a choice of a minimum Kolmogorov complexity function in $S_{f_c}^1$ for each function in $S_f^2$. For example, if $S_f^1$ contains all linear functions and $S_f^2$ is the class of all polynomials then we can view a monotonic map $\mathbf{F_B} \to \mathbf{F_C}$ as selecting a linear approximation for each polynomial in $S_f^2$.

**Proposition A.10.** *For some function $g \in S_f^2$ define its minimal S-overapproximation to be the function $h \in S_{f_c}^1$ where $g \leq_S h$ and $\forall h' \in S_{f_c}^1$ where $g \leq_S h'$ we have $h \leq_S h'$. If this function exists it is unique.*

*Proof.* Suppose $h_1, h_2$ are both minimal $S$-overapproximations of $g$. Then $h_1 \leq_S h_2$ and $h_2 \leq_S h_1$ which by the definition of $S_{f_c}^1$ implies that $h_1 = h_2$. $\square$

**Proposition A.11.** *For some function $g \in S_f^2$ define its maximal S-underapproximation to be the function $h \in S_{f_c}^1$ where $h \leq_S g$ and $\forall h' \in S_{f_c}^1$ where $h' \leq_S g$ we have $h' \leq_S h$. If this function exists it is unique.*

*Proof.* Suppose $h_1, h_2$ are both maximal $S$-underapproximations of $g$. Then $h_2 \leq_S h_1$ and $h_1 \leq_S h_2$ which by the definition of $S_{f_c}^1$ implies that $h_1 = h_2$. $\square$

**Proposition A.12.** *Suppose that for some $g \in S_f^2$ there exists some $h \in S_{f_c}^1$ such that $h =_S g$. Then $h$ will be both the minimal S-overapproximation and the maximal S-underapproximation of $g$.*

*Proof.* To start, note that $h$ must satisfy $g \leq_S h$ and for any $h' \in S_{f_c}^1$ where $g \leq_S h'$ we have:

$$h =_S g \leq_S h'$$

so $h$ is the minimal $S$-overapproximation of $g$.

Next, note that $h$ must satisfy $h \leq_S g$ and for any $h' \in S_{f_c}^1$ where $h' \leq_S g$ we have:

$$h' \leq_S g =_S h$$

so $h$ is also the maximal $S$-underapproximation of $g$. $\square$

We can now show the following:

**Proposition A.13.** *Define both $K : \mathbf{F_A} \hookrightarrow \mathbf{F_C}$ and $G : \mathbf{F_A} \hookrightarrow \mathbf{F_B}$ to be inclusion maps. Then:*

- *Suppose that for any function $g \in S_f^2$ there exists a minimal S-overapproximation $h$ of $g$. Then the left Kan extension of $K$ along $G$ is the monotonic map $Lan_G K$ that maps $g$ to $h$.*

- *Suppose that for any function $g \in S_f^2$ there exists a maximal S-underapproximation $h$ of $g$. Then the right Kan extension of $K$ along $G$ is the monotonic map $Ran_G K$ that maps $g$ to $h$.*

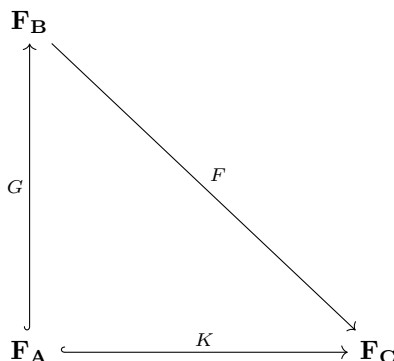

*Proof.* We first show that $Lan_G K$ is monotonic when it exists. Since $\mathbf{F_B}, \mathbf{F_C}$ are preorders we simply need to show that when $f_1 \leq_S f_2$ then $Lan_G K(f_1) \leq_S Lan_G K(f_2)$. Since $f_2 \leq_S Lan_G K(f_2)$ by the definition of the minimal $S$-overapproximation of $f_2$ we have that $f_1 \leq_S Lan_G K(f_2)$. Then $Lan_G K(f_1) \leq_S Lan_G K(f_2)$ by the definition of the minimal $S$-overapproximation of $f_1$.

We next show that $Ran_G K$ is monotonic when it exists. Since $\mathbf{F_B}, \mathbf{F_C}$ are preorders we simply need to show that when $f_1 \leq_S f_2$ then $Ran_G K(f_1) \leq_S Ran_G K(f_2)$. Since $Ran_G K(f_1) \leq_S f_1$ by the definition of the maximal $S$-underapproximation of $f_1$ we have that $Ran_G K(f_1) \leq_S f_2$. Then $Ran_G K(f_1) \leq_S Ran_G K(f_2)$ by the definition of the maximal $S$-underapproximation of $f_2$.

Next, we will show that $Lan_G K$ and $Ran_G K$ are respectively the left and right Kan extensions when they exist. First, by Proposition A.12 if $f \in S^1_{f_c}$ then $f$ must be both the minimal $S$-overapproximation and maximal $S$-underapproximation of $f$. Therefore we have:

$$K(f) = Lan_G K(f) = Ran_G K(f)$$

Next, consider any monotonic map $M_L : \mathbf{F_B} \to \mathbf{F_C}$ such that $\forall f \in S^1_{f_c}, K(f) \leq_S M_L(f)$. Since $f =_S K(f)$ this implies $f \leq_S M_L(f)$ so by the definition of the minimal $S$-overapproximation $Lan_G K(f) \leq_S M_L(f)$.

Next, consider any monotonic map $M_R : \mathbf{F_B} \to \mathbf{F_C}$ such that $\forall f \in S^1_{f_c}, M_R(f) \leq_S K(f)$. Since $K(f) =_S f$ this implies $M_R(f) \leq_S f$ so by the definition of the maximal $S$-underapproximation $M_R(f) \leq Ran_G K(f)$.
□

Intuitively, the Kan extensions of the inclusion map $K : \mathbf{F_A} \to \mathbf{F_C}$ along the inclusion map $G : \mathbf{F_A} \to \mathbf{F_B}$ map a function $g \in S^2_f$ to its best $S^1_f$-approximations over the points in $S$.

For example, suppose $I = O = \mathbb{R}$, $g$ is a polynomial, $S^1_f$ is the set of lines defined by all pairs of points in $S$ and $S^2_f = S^1_f \cup g$. $Lan_G K$ and $Ran_G K$ may or may not exist depending on the choice of $S$ and $g$. In Figure 3 we give an example $S, g$ in which $Lan_G K$ exists and $Ran_G K$ does not (left) and an example $S, g$ in which $Ran_G K$ exists and $Lan_G K$ does not (right).

As another example, suppose $I = O = \mathbb{R}$, $S^1_f$ is a subset of all polynomials of degree $|S| - 1$ and $S^2_f$ is a subset of all functions $\mathbb{R} \to \mathbb{R}$. Since there always exists a unique $n - 1$ degree polynomial through $n$ unique points, for any $S$ there exists some $S^1_f$ so that both $Lan_G K$ and $Ran_G K$ exist and map $g \in S^2_f$ to the unique $|S| - 1$ degree polynomial that passes through the points $\{(x, g(x)) \mid x \in S\}$.

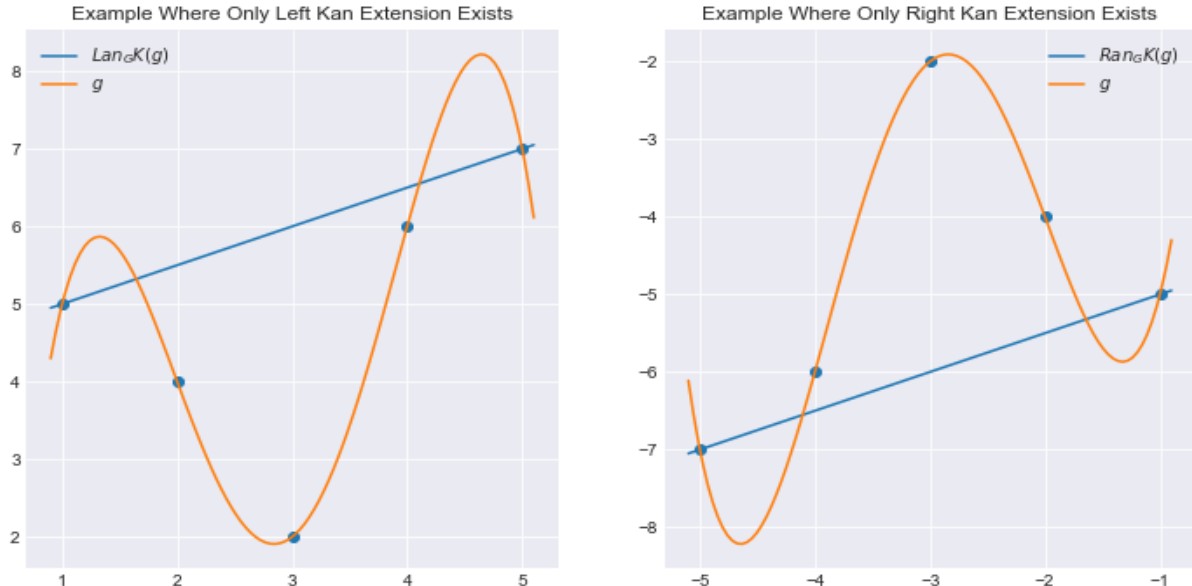

**Figure 3:** *Left and right Kan extensions of* $K : \mathbf{F_A} \hookrightarrow \mathbf{F_C}$ *along* $G : \mathbf{F_A} \hookrightarrow \mathbf{F_B}$ *for two example sets* $S$ *and polynomials* $g$ *where* $S_f^1$ *is the class of lines and* $S_f^2 = S_f^1 \cup g$.

