# OpenReview forum: "Learning with Kan Extensions"
_TMLR — Rejected by TMLR_

### Review · Reviewer_meGc · 2022-08-05

**Summary Of Contributions:**

This paper explains how right and left Kan extensions, tools issued from category theory, can be used in a machine learning setting.

Despite being rather unfamiliar with category theory, I found the paper to be quite an interesting read. For the same reason and to hand out my review in time (and before my own vacations), I could only perform a superficial check of the mathematics of the paper, but those are correct as far as I can tell (nevertheless, they should be checked by an expert of the field).

The paper is very well-written and pleasant to read, with examples helping to grasp concepts expressed in a mathematically abstract way. Some experiments are led to show how those concepts can be applied, with quite satisfying results.



**Broader Impact Concerns:**

No concerns

**Requested Changes:**

The paper is quite nice as it is, and I would not especially change anything. I have a couple of questions that authors may wish to answer here (and maybe also in the paper, but I do not consider it necessary):

* Since Kan extensions heavily rely on the concept of ordered spaces and mapping between them, I was wondering if it would not be an ideal tool to use in the context of monotonic classification, where one assumes monotonic relationships between the attributes and the resulting (ordered) class? See for example "Monotonic classification: An overview on algorithms, performance measures and data sets" for a recent review.

* Similarly, I am wondering how easy it would be to extend the proposal made for classification to the case where the set of classes is unordered? Should one impose an order/permutation, or proceed pairwisely?

* Regarding the first experiment, what would be the results if we rejected all those samples for which the two extensions disagree? Said otherwise, is disagreement between Kan extensions a useful tool to implement a (partial) reject option?

* Another connection that would seem possible to make is with the notion of rough sets, and the different regions they can induce in the sample/class space? The two notions seem indeed related. A quick search also show some papers trying to connect the two notions, but those appear to be rare (e.g., https://ieeexplore.ieee.org/document/4666018/)

* Finally, regarding the clustering case, it seems the approach can well handle "must-link" constraints when we learn that two objects should be together, but I was wondering if it could handle "cannot-link" constraints, i.e., specifying that two objects cannot be together in the resulting clustering and guaranteeing that this will be the case in the right and left extensions.

**Strengths And Weaknesses:**

+: an interesting take on learning problems, connecting them to category theory
+: a very well-written paper
+: examples and first experiments demonstrating the applicability of the ideas and their potential usefulness

No real perceived weaknesses, but my expertise of the used mathematics is limited.

---

> ### Author Response · Authors · 2022-08-08
> **Thank you for your helpful review**
>
> Thank you for your helpful review. We are really glad that you found this paper interesting.
>
> * Here is one way that we could adapt this to work for monotonic classification. Assume we have a classification problem with N features, M of which require a monotonicity constraint. We can modify the left and right Kan classifiers that we describe in Section 3 to solve this problem. Rather than apply a transformation function $f: R^N \rightarrow R^a$ that minimizes the ordering loss (Definition 3.2), we can apply a transformation function $f': R^N \rightarrow R^{M+a}$ that keeps the $M$ features that require monotonicity constant and only minimizes the ordering loss over transformations of the remaining $N-M$ features.
> * There is no obvious way to extend the left and right Kan classifiers to work when there is no ordering on the target classes. However, if we can impose some kind of preorder on the target classes then we can pretty easily extend these algorithms to generate outputs in this preorder rather than $\\{false, true\\}$. For a large set of unordered classes there might be an interesting ensemble algorithm where we generate a bunch of random preorders, run the algorithm for each preorder, and take the average of the predictions. This would probably end up behaving like a nearest neighbor classifier
> * The left Kan classifier has no false negatives on the training set and the right Kan classifier has no false positives on the training set. Therefore the region where they agree has perfect accuracy on the training set. The size of this region will depend on our choice of transformation function $f$. By Proposition 3.3 the disagreement region is the empty set when this transformation function drives the ordering loss to 0. If we want to allow for a disagreement region within which we reject samples we can use a lower complexity transformation function $f$ that lacks the capacity to minimize the ordering loss but may be less prone to overfitting.
> * We have not considered linking rough sets and Kan extensions. One potential connection would be to follow a similar path to the one described in [Functorial Manifold Learning](https://www.cl.cam.ac.uk/events/act2021/papers/ACT_2021_paper_1.pdf). Essentially, we can represent the sliding scale of element inclusion in a fuzzy set as a functor (a generalization of a monotonic map).
> * Yes, this construction handles this case. Suppose the metric space $(X, d_X)$ is in $T$, the points $x_1,x_2$ are in $X$, and $K(X, d_X)$ is a partition of $X$ in which $x_1,x_2$ are in different clusters. Suppose also that $(X, d'_X)$ is a metric space in $D$ and $(X, d'_X) \leq (X, d_X)$. Then any monotonic map from $D$ to $Part$ (including both Kan extensions) must map $(X, d'_X)$ to a partition of $X$ in which $x_1,x_2$ are in different clusters.

---

### Review · Reviewer_7opt · 2022-08-11

**Summary Of Contributions:**

I have to state that I know nothing about the category theory that the authors are using in this paper. So I only understand very little about this paper.

If I represent a regular ML researcher, this paper does not seem to be a good fit to TMLR because it has used too much mathematical content beyond the knowledge of a regular ML researcher.
The authors should consider submitting it to an applied math journal or an ML journal and provide a long and detailed introduction on the math.

This paper introduces the Kan extension to the classification and clustering problem.


**Broader Impact Concerns:**

This paper does not include a statement on the broader impact. And I cannot understand most part of the paper so I cannot judge its broader impact.

**Requested Changes:**

1. I will recommend the authors to resubmit this paper to an applied math journal. The math concepts are interesting but unclear to me how they can be useful in ML problems.

2. Provide more explanations and improve the writing. The authors have to improve the writing and provide more explanations so that regular ML researchers can understand the contents. For instance, how will a classifier be represented in the diagram (I think a classifier will represent a map)? and how can we think of the map between other two pre-order sets?

3. Provide a clear motivation. As is mentioned in the weakness, there is no clear motivation on why we need the mathematics introduced in this paper. It seems to me that this idea is just very cool mathematical concepts but not clear to me if it will be useful in practice.

4. Provide a clear explanation on how to use the method. There is no clear explanation on how to use this extension. Do we need to start with a classifier? How do we construct another map between pre-order sets? Does this related to feature transformation derived from kernel methods?

**Strengths And Weaknesses:**

+ New math concept.

- Very poor explanation. The writing is so bad that most ML researchers will not be able to understand it. There are too many math concepts that are not in the usual ML curriculum.
- Lack of motivation. Why do we want to use Kan extension? Does it provide improvement on the performance? Does it show connections among different methods? The motivation of this paper is totally unclear to me.
- Unclear how to use the proposed method. There is no clear explanation how to construct the Kan extension from a given classifier or clustering algorithm.
- Limitation of the method. The idea relies heavily on the pre-order. However, feature space often does not have a pre-order, especially when there are more than 1 variable

---

### Review · Reviewer_WqRC · 2022-08-13

**Summary Of Contributions:**

The paper proposes ways of using Kan extensions to functionally define methods
for a few learning tasks, in particular varieties of classification tasks. The
first task is simply binary classification with real-valued features, and the
classifier has the form of a threshold on some (unspecified) norm of a linear
transformation of the instance's features. In each case, there are two
classifiers (obtained from the "left" and "right" Kan extensions), that are
respectively biased towards positive and negative classification. The second
task is essentially a multi-class classification problem where various sets of
points lie in various metric spaces (i.e., are labeled by various distances that
obey the triangle inequality, etc.) -- it is framed as a clustering problem, but
the objective does not (seem to) have much to do with the geometry of the
points, but rather recovering a specified set of partitions that we can view as
class labels.

Some small experiments on Fashion MNIST demonstrate that the proposed
classifiers obtain nontrivial (but surely not state-of-the-art) performance on
these tasks.


**Broader Impact Concerns:**

No concerns.

**Requested Changes:**

I am confused about a few key points in the proposed (example) constructions.
First, what norm is being used by the classifier in the first, binary
classification task? Second, what role do the collection of nonexpansive maps
play in the second task? These must be clarified in the text.

Some relatex work is missing, and consequently the discussion of the Kan
extension approach versus "traditional machine learning" borders on factually
incorrect: learning tasks of precisely the form "minimize false positives
subject to no false negatives on some set" have been considered previously,
under various names. See for example the "heuristic learning" task considered
by Pitt and Valiant [1] and "positive/negative reliable learning" as considered
by Kalai et al. [2]. The work should note that the Kan extension approach solves
tasks in such models (and must not suggest that this is a novel formulation).

Ideally the work would compare to the existing algorithms for reliable/heuristic
learning in the experiments, especially the first classification experiment that
currently has no baseline (except, e.g., "trivial performance").

[1] L. Pitt and L. G. Valiant. Computational limitations on learning from examples. Journal of the ACM (JACM), 35(4):965-984, 1988.

[2] A. T. Kalai, V. Kanade, and Y. Mansour. Reliable agnostic learning. Journal of Computer and System Sciences, 78(5):1481-1495, 2012.


**Strengths And Weaknesses:**

The main strength of the work is the unusual perspective it takes. As far as I
am aware, the proposed approach to obtaining classifiers via Kan extensions is
novel. I have not seen these notions used in machine learning previously.

There are a handful of weaknesses, however: most critically, the paper is
unclear on some key points and misses some very relevant prior work (see the
requested changes, below).

The experiments are somewhat weak, especially the first experiment that does not
even include a baseline.

I also generally found the formulation of the "clustering" task confusing. It is
very strange to me that the resulting "clustering" map acts on metric spaces and
produces partitions. I'd intuitively have expected it to map points from Ob(D)
to an index for some part of the partition of Ob(D) (or simply output a
partition rather than a partition-valued mapping).

Finally, it seems a bit piecemeal: we can use Kan extensions to obtain
constructions of classifiers for some tasks, but it isn't clear why these
classifiers would be desirable in any particular way. Also, In each case, it
appears that some ad hoc choices were made in the constructions, e.g., the use
of UMAP to induce a metric on Fashion MNIST. I would view it more favorably if
there were some general principles that defined the classifiers, indicated when
and why this approach would be likely to be useful, etc.

---

### Decision · Action_Editors · 2022-09-25

**Recommendation:** Reject

**Comment:**

The paper proposes a potentially interesting point of view connecting the Kan extension to machine learning. However, the reviewers found several issues, including clarity, connections to related work, and comparison to appropriate baselines, that should be addressed. The authors ignored most of the feedback, and did not explain how they would correct these issues. Therefore, the submission is rejected. The authors are encouraged to address the issues highlighted by the reviewers and resubmit.